# On the Hardness of Online Nonconvex Optimization with Single Oracle Feedback

**Ziwei Guan**[§], **Yi Zhou**[‡] **& Yingbin Liang**[§]

Department of Electrical and Computer Engineering, Ohio State University[§]
Department of Electrical and Computer Engineering, University of Utah[‡]
{guan.283, liang.889}@osu.edu, yi.zhou@utah.edu

## Abstract

Online nonconvex optimization has been an active area of research recently. Previous studies either considered the global regret with full information about the objective functions, or studied the local regret with window-smoothed objective functions, which required access to unlimited number of gradient oracles per time step. In this paper, we focus on the more challenging and practical setting, where access to only a single oracle is allowed per time step, and take the local regret of the original (i.e., unsmoothed) objective functions as the performance metric. Specifically, for both settings respectively with a single exact and stochastic gradient oracle feedback, we derive lower bounds on the local regret and show that the classical online (stochastic) gradient descent algorithms are optimal in the class of linear-span algorithms. Moreover, for the more challenging setting with a single function value oracle feedback, we develop an online algorithm based on a one-point running difference gradient estimator, and show that such an algorithm achieves a local regret that a generic stochastic gradient oracle can best achieve.

## 1 Introduction

Online optimization is a popular learning framework that models sequential decision-making in a non-stationary environment, and it has attracted a lot of attention in the past decade (Hazan et al., 2016; Orabona, 2019). In online optimization, an agent interacts with the environment by selecting a sequence of control variables $x_1, \ldots, x_t$ and receiving information about loss functions $f_1(\cdot), \ldots, f_t(\cdot)$ from the environment. Specifically, at each iteration $t$, the agent selects a control variable $x_t$ based on the historical information about $f_1, \ldots, f_{t-1}$. After the selection has been made, certain feedback about $f_t$ is revealed to the agent, who then selects the next control variable $x_{t+1}$ based on the updated information. The goal of the agent is to select $x_t$ sequentially to minimize a certain metric of regret, typically defined as the cumulative difference between the function value at the chosen control variable and the minimal function value with hindsight information.

In the existing studies on online optimization, many efficient online algorithms have been proposed to minimize different types of regret for **convex** objective functions, whose global minimum can be efficiently achieved with gradient oracles (Agarwal et al., 2010; Suggala & Netrapalli, 2020; Héliou et al., 2020; Zinkevich, 2003; Hall & Willett, 2013; Jadbabaie et al., 2015; Mokhtari et al., 2016; Zhang et al., 2018). As a comparison, the more challenging online **nonconvex** optimization has not been well explored yet. Specifically, the existing studies on online nonconvex optimization have focused on two major settings. One line of work adopted the *global regret* as the performance metric (Krichene et al., 2015; Agarwal et al., 2019; Lesage-Landry et al., 2020; Héliou et al., 2020), which compares the algorithm output to the global minimum of the nonconvex objective functions. However, accessing the global minimum of nonconvex functions is typically infeasible. Another line of work considered the more feasible *local regret* (Hazan et al., 2017; Aydore et al., 2019; Hallak et al., 2021; Guan et al., 2023), which compares the control variables to the stationary points of corresponding nonconvex functions. Importantly, these nonconvex functions are smoothed versions of the online nonconvex objective functions $f_t$ averaged over a sliding window. These studies designed online algorithms that achieve a sublinear local regret of the smoothed nonconvex objective functions with a sufficiently large window length. However, the local regret of window-smoothed functions can be very different from that of the original online objective functions and hence may not accurately reflect

the actual performance of the system, especially when the functions $f_t$ change rapidly. Therefore, we are motivated to study the following fundamental problems in this work.

- *Q1: What are the fundamental lower bounds on the local regret (without function smoothing) for online nonconvex optimization with different types of single oracle feedback, e.g., exact gradient, stochastic gradient, and function value oracles?*

- *Q2: What online learning algorithms can achieve these lower bounds on the local regret using only single oracle feedback of nonconvex functions?*

## 1.1 OUR CONTRIBUTIONS

In this paper, we follow the line of research on the local regret of online *nonconvex* optimization, and focus on the setting with only a **single** oracle feedback per time step. In particular, we study the local regret of the original online objective functions (i.e., without window-smoothing). Below we discuss the major challenges in each setting and summarize our contributions.

- **Single Gradient Oracle (SGO) feedback,** where only one gradient oracle is available at a time. In this setting, instantiating the existing analysis (Hazan et al., 2017; Guan et al., 2023) by taking the window size $w = 1$ provides an $\Omega(T)$ lower bound on the local regret, which ignores the intrinsic function variation over time and can be overly pessimistic when the functions change slowly. We provide a tighter and problem-dependent lower bound on the local regret that captures how the function variation affects the hardness of the problem. Then, we show that the classical online gradient descent algorithm achieves the optimal regret bound for the class of linear-span algorithms.

- **Single Stochastic Gradient Oracle (SSGO) feedback**, where only one stochastic gradient oracle is available at a time and the local regret is harder to optimize. In this setting, the existing study (Guan et al., 2023) provides an $\Omega(T)$ lower bound by taking window size $w = 1$, but it does not capture the impact of function variation and can be overly pessimistic. We characterize the challenge through establishing another tight lower bound on the expected local regret. Then, we show that the online stochastic gradient descent algorithm achieves the optimal regret for the class of linear-span algorithms.

- **Single function value oracle (SVO) feedback**, where only one function value oracle is available at a time. In this setting, the challenge lies in constructing a good gradient estimator with only one function value oracle.We found that online gradient descent with the classical one-point gradient estimator has poor performance due to large variance. To address this, we developed a one-point running difference gradient estimator. We show that such a zeroth-order gradient estimator, when applied to the online gradient descent algorithm, achieves an optimal local regret for the class of linear-span algorithms with generic stochastic gradient oracle.

The above three settings capture different application scenarios. The SGO setting applies to white-box systems with known objective functions, where exact gradients can be calculated. However, many systems can have various uncertainties, e.g., loss functions depending on random samples or systems with intrinsic noise. In such a case, even if the system objective function is known, the calculation of their gradients can still be stochastic, leading to the SSGO setting. Moreover, the SVO setting is suitable to model black-box systems, where objective functions are unknown but only function values can be queried. For example, the recommendation system can provide users' ratings (i.e., reward values), but the reward function following which users provide ratings is typically unknown.

## 1.2 RELATED WORK

**Online convex optimization:** Online convex optimization has been extensively studied in the past, and we refer the readers to the standard textbooks (Hazan et al., 2016; Orabona, 2019; Shalev-Shwartz, 2012) and a recent survey (Hoi et al., 2021) to obtain a comprehensive understanding. Below we summarize a few main directions on the topic.

Studies on online convex optimization can be generally divided into three categories based on which notion of regret they are interested in and what feedback information a learner has for the design. Extensive works such as Agarwal et al. (2010); Suggala & Netrapalli (2020); Héliou et al. (2020) studied static regret, which is defined as the difference between cumulative losses and the minimum cumulative losses with hindsight information. Many other works (Zinkevich, 2003; Hall & Willett, 2013; Jadbabaie et al., 2015; Mokhtari et al., 2016; Zhang et al., 2018; Héliou et al., 2020) studied

the dynamic regret, where the comparison is made to the function value at a given reference point (typically chosen as a global minimum of the instantaneous objective function) at each time step. Among these three lines of research, the studies on the dynamic regret (Zinkevich, 2003; Hall & Willett, 2013; Jadbabaie et al., 2015; Mokhtari et al., 2016; Zhang et al., 2018; Héliou et al., 2020) are closest to this paper. Their development of upper bounds on the dynamic regret uses the Lipschitz smoothness, gradient update direction, and the convexity of online functions. In particular, their upper bounds involve various types of function variation such as the path length (defined as the summation of differences between two adjacent global minimal variables) and the gradient variation. In our study of online nonconvex optimization, the development of upper bounds on the dynamic regret also uses the Lipschitz smoothness and the gradient update direction (which are the same as online convex optimization). However, since there is no convexity property we can use here, we leverage the Lipschitz smoothness of each objective function to connect the local regret to the cumulative difference in function values, which leads to a different function variation we define in eq. (2).

Further, a number of works such as Hazan & Seshadhri (2009); Daniely et al. (2015); Zhang et al. (2019); Garber & Kretzu (2022) studied the adaptive regret, which is defined as the static regret on the subsets of all iterations, bridging between the dynamic regret and the static regret. Online convex optimization has also been studied under function value feedback, which captures many practical applications. In particular, Flaxman et al. (2005) and Saha & Tewari (2011) studied the setting with one-point gradient estimators, and Agarwal et al. (2010) studied the setting with multiple-point estimators. Further, Cao & Liu (2018); Kim & Lee (2023) studied functional constrained online optimization and Yi et al. (2020); Wang et al. (2022) studied distributed online optimization with both one-point and two-point estimators.

**Online nonconvex optimization:** One line of studies on online nonconvex optimization considered the *global regret* as the performance metric, which is defined as the cumulative discrepancy between the loss of the algorithm output and the lowest possible loss. Particularly, Krichene et al. (2015) extended the Hedge algorithm to the continuum and showed it achieves a sublinear regret. Yang et al. (2018) developed an improved algorithm based on the Hedge algorithm with a novel weighting strategy. Agarwal et al. (2019) extends the bandit algorithm of follow-the-perturbed-leader to the nonconvex setting, and Suggala & Netrapalli (2020) further showed that such an algorithm achieves the optimal regret. Online nonconvex optimization has also been studied under certain assumptions on nonconvex objective functions, such as the weak pseudo-convex condition (Gao et al., 2018) and the Polyak-Lojasiewicz condition (Mulvaney-Kemp et al., 2022). Moreover, the global regret minimization of online nonconvex functions has also been studied under the *function value feedback* in Gao et al. (2018); Héliou et al. (2020; 2021); Gao et al. (2018).

Another line of research studied the more feasible *local regret* that compares the control variables to the stationary points of window-smoothed nonconvex functions. Particularly, Hazan et al. (2017) provided the first analysis, where the objective functions are smoothed via a sliding window and developed a lower bound on the local regret. The authors further devised a nested-loop algorithm and showed that it achieves the regret lower bound. Aydore et al. (2019) adopted the nested-loop type algorithm to a dynamic environment application, and Hallak et al. (2021) adopted the same approach and extended it to the setting with nonsmooth functions. Guan et al. (2023) considered the setting with a limited number of feedback concerning the gradient or the function value of the window-smoothed objectives and provided tight lower and upper bounds for linear-span algorithms. Our study follows this line of research with the following main differences. First, we do not take window-smoothed objective functions but rather focus on the original functions and their corresponding local regret and provide a problem-dependent analysis based on the function variation. Second, all the algorithms developed in the above studies require multiple feedback oracles concerning $f_t$ due to the window-smoothing, whereas we study the settings with only a single oracle feedback of $f_t$. It is worth noting that the aforementioned previous studies on the complimentary regret metric based on window-smoothed functions are still valuable, particularly for studying slowly changing environments. In the future, it is interesting to continue their line of work and obtain refined problem-dependent bounds based on function variations.

**Gradient estimation with function value oracles:** Various gradient estimation methods have been applied to online optimization when there is only function value (i.e., bandit) feedback. The performance of these gradient estimators highly affect the performance of their corresponding online algorithms. Specifically, the conventional one-point gradient estimator was developed in Flaxman et al. (2005); Dekel et al. (2015); Gasnikov et al. (2017) to estimate the gradient of a function based only

on one function value. Two-point gradient estimators (Agarwal et al., 2010; Nesterov & Spokoiny, 2017; Shamir, 2017) are also designed to estimate the gradient based on two function values and can achieve much better performance over the conventional one-point estimator. Recently, Zhang et al. (2022) proposed a novel one-point estimator which uses an immediate previous function value during iterations to serve as the second function value in the gradient estimator to reduce the variance. Such an estimator has been shown in Zhang et al. (2022) to achieve a much better performance than the conventional one-point estimator but still performs weaker than the two-point estimator. Inspired by such an estimator, in this paper, we propose a one-point running difference estimator for online nonconvex optimization and show that it enjoys the same regret bound as two-point estimator.

## 2 PRELIMINARY

In this paper, we consider the online nonconvex optimization (ONO) problem with a sequence of objective functions $f_1, f_2, \ldots, f_t, \ldots$. The objective functions are smooth and bounded on $\mathbb{R}^d$ and are generally nonconvex. We adopt the following standard assumption that is widely used in the online optimization literature (Agarwal et al., 2010; Flaxman et al., 2005; Hazan et al., 2017; Aydore et al., 2019; Zhao et al., 2020; Hallak et al., 2021).

**Assumption 1.** *For all $t \geq 1$, function $f_t : \mathbb{R}^d \to \mathbb{R}$ satisfies $|f_t(x)| \leq M$ for some $M \geq 0$ and for all $x \in \mathbb{R}^d$. Moreover, there exist universal constants $L_0, L_1 \geq 0$ such that for any $x, y \in \mathbb{R}^d$, $|f(x) - f(y)| \leq L_0 \|x - y\|$, and $\|\nabla f(x) - \nabla f(y)\| \leq L_1 \|x - y\|$, where $\|\cdot\|$ denotes the $\ell_2$ norm.*

At each iteration $t$, the agent first submits a control variable $x_t \in \mathbb{R}^d$ to the system based on the historical feedback. After that, the feedback about objective function $f_t$ is revealed from the system, and based on that, the agent further determines the next control variable. For such an ONO problem, we study the following local regret.

$$\text{(Local regret):} \quad \mathfrak{R}(T) := \sum_{t=1}^{T} \|\nabla f_t(x_t)\|^2. \tag{1}$$

Intuitively, the local regret defined in eq. (1) compares the control variables to the stationary points $x_t^*$ that satisfies $\nabla f_t(x_t^*) = \mathbf{0}$. It can also be interpreted as the dynamic local regret because the comparison baseline is a stationary point of the instantaneous objective function $f_t$.

We note that the previous work of online nonconvex optimization (Hazan et al., 2017; Hallak et al., 2021; Guan et al., 2023) studied the local regret with window-smoothing, which is defined as $\mathfrak{R}_w(T) := \sum_{t=1}^{T} \|\nabla F_{t,w}(x_t)\|$, where $F_{t,w}(x) := \frac{1}{w} \sum_{i=t-w+1}^{t} f_i(x)$ corresponds to the smoothed objective function averaged over a sliding window of size $w$. It reduces to our definition of the local regret in eq. (1) when $w = 1$. In Hazan et al. (2017); Hallak et al. (2021); Guan et al. (2023), the window size $w$ is chosen to be large so that the smoothed functions $\{F_{t,w}\}_t$ change slowly over the iterations, making it easier to minimize the local regret. However, the local regret of window-smoothed functions can be very different from that of the original functions and hence may not accurately reflect the actual performance of the system, particularly when functions change rapidly.

In this work, we focus on online nonconvex optimization with *single oracle feedback* about $f_t$ per iteration, and our goal is to develop algorithms that minimize the local regret in eq. (1). In particular, we consider the following single oracle feedback settings:

- **Single Gradient Oracle (SGO) feedback:** At each time $t$, one can only access a single exact gradient oracle.

- **Single Stochastic Gradient Oracle (SSGO) feedback:** At each time $t$, one can only access a single stochastic gradient oracle.

- **Single Value Oracle (SVO) feedback:** At each time $t$, one can only access a single function value oracle.

We also define the following notion of **function variation** over time, a fundamental quantity that is useful to characterize the local regret in the nonconvex setting.

$$V_T := \sum_{t=2}^{T+1} \sup_{x \in \mathbb{R}^d} |f_{t-1}(x) - f_t(x)|. \tag{2}$$

The above function variation intuitively reflects how much the online learning objective functions vary over time. In particular, when $V_T = 0$, all $f_t$'s are the same as $f_1$, and online learning problem

reduces to an offline optimization problem. On the other hand, when $V_T = cT$ for some constant $c > 0$, the number of objective functions $f_t$ that are totally different from all previous ones can scale linearly with $T$. No algorithm can perform well for these rapidly changing functions $f_t$ because all received information is about $f_1$ up to $f_{t-1}$, which contains little information about $f_t$.

## 3 ONLINE NONCONVEX OPTIMIZATION WITH SGO

Previous studies (Hazan et al., 2017; Hallak et al., 2021; Guan et al., 2023) of online *nonconvex* optimization have been focusing on window-smoothed functions that require access to multiple gradient oracles of $f_t$. In this section, we study online nonconvex optimization with SGO, where only a *single* access to the gradient oracle of $f_t$ is available at each time $t$.

### 3.1 LOWER BOUND ON LOCAL REGRET UNDER SGO

In order to understand the complexity limit of the problem under SGO, we establish a lower bound on the local regret. We first specify the algorithm class that we consider as follows.

**Definition 1** (Linear-span (Nesterov, 2003)). *The class of online learning algorithms $\mathcal{A}$ with SGO feedback at each time generates a sequence of variables $\{x_t\}_{t=1}^{\infty}$ according to*

$$x_{t+1} \in \left\{ x_1 + \sum_{i=1}^{t} a_{t,i} \nabla f_i(x_i) : a_{t,i} \in \mathbb{R}, i = 1, \ldots, t \right\}.$$

The above class of algorithms queries only a single gradient oracle each time step, and it covers many widely used gradient-based algorithms, including gradient descent, accelerated gradient descent, etc.

The following theorem establishes the first lower bound on the local regret in online nonconvex optimization for all linear-span algorithms with SGO feedback defined in Definition 1.

**Theorem 1.** *Consider any algorithm $\mathcal{A}$ that satisfies Definition 1 and has access to a single gradient oracle $\nabla f_t$ at each iteration. Then, for problems with the dimension $d \geq \Omega(1 + V_T)$, there exist $\{f_t\}_{t=1}^{\infty}$ with function variation $V_T$ (defined in eq. (2)) and satisfying Assumption 1, for which $\Re(T) \geq \Omega(1 + V_T)$.*

Theorem 1 shows that the hardness of the problem over any linear-span algorithm is entirely determined by the function variation defined in eq. (2), i.e., the changes of the objective functions. If the objective function $f_t$ is totally different from the previous functions, then all previous learning history is not helpful, and the agent is forced to learn each $f_t$ from scratch. In such a case, the local regret must be $\Omega(T)$. Nonetheless, if the changes of functions are small and scale sublinearly with $T$, it can be possible to achieve a sublinear regret.

As a comparison, although the existing works (Hazan et al., 2017; Guan et al., 2023) provided lower bounds on the local regret that can instantiate to our setting of SGO (Hazan et al. (2017, Theorem 2.7) with $w = 1$ and Guan et al. (2023, Theorem 1) with $w = 1$ and $\sigma = 0$), their resulting lower bounds do not capture the impact of the function variation $V_T$. They are vacuous with the order of $\Omega(T)$. In contrast, our lower bound in Theorem 1 is tighter and problem-dependent by capturing how the function variation affects the hardness of the problem.

*Proof Outline of Theorem 1.* The details of the proof can be found in Appendix A.1. The main idea of the proof is as follows. Given a total budget of $V_T$ on the function variation, the objective function can have $\Omega(1 + V_T)$ times of rapid change. Thus, we divide the total time steps into $\Omega(1 + V_T)$ blocks, choose the same objective function within each block and change the objective function across blocks. We then construct a series of functions whose gradients are orthogonal to each other and assign them to these blocks, which hinders the agent from constructing $\nabla f_t(x_t)$ based on feedback from previous blocks. This construction of $\{f_t\}_{t=1}^{T}$ forces the agent to restart the learning process in each block. The agent suffers from a high value of $\|\nabla f_t(x_t)\|^2$ at the beginning of each restart, and thus the local regret is doom to be $\Omega(1 + V_T)$. $\square$

### 3.2 LOCAL REGRET OF ONLINE GRADIENT DESCENT

Consider the simple online gradient descent (OGD) algorithm described in Algorithm 1, which clearly belongs to the class of linear-span algorithms defined in Definition 1. At each iteration $t + 1$, the OGD algorithm with constant stepsize $\eta > 0$ takes the update

$$x_{t+1} = x_t - \eta \nabla f_t(x_t). \tag{3}$$

---

**Algorithm 1** Online Gradient Descent (OGD)

---

**Input:** Initial point $x_1$, stepsizes $\eta$
**for** $t = 1, \ldots, T$ **do**
    Update $x_{t+1}$ based on eq. (3)
**end for**

---

Our next result shows that OGD achieves the local regret $\mathcal{O}(1 + V_T)$ for online nonconvex optimization, which matches the lower bound in Theorem 1. This implies that OGD is optimal in the class of linear-span algorithms for online nonconvex optimization with SGO.

**Theorem 2.** *Suppose Assumption 1 holds. Consider Algorithm 1 with initial point $x_1 = 0$, and stepsize $\eta = \frac{1}{L_1}$. Then, we have $\mathfrak{R}(T) \leq \mathcal{O}(1 + V_T)$.*

The proof of Theorem 2 is included in Appendix A.2. The main idea is to leverage the Lipschitz smoothness of each objective function to connect the local regret to the cumulative difference $\sum_{t=1}^{T} f_{t+1}(x_{t+1}) - f_t(x_{t+1})$ that is upper-bounded by $V_T$.

## 4 ONLINE NONCONVEX OPTIMIZATION WITH SSGO

In many practical online learning scenarios, the exact gradient feedback may not be available, and one has access to only noisy stochastic gradient oracles. Thus, in this section, we consider the setting where only a single stochastic gradient oracle (SSGO) is available at each time step. We adopt the following standard unbiasedness and bounded variance assumptions on the stochastic gradient, which is widely adopted in the online optimization literature (Agarwal et al., 2010; Flaxman et al., 2005; Hallak et al., 2021; Hazan et al., 2017; Guan et al., 2023).

**Assumption 2.** *For every $t$, the stochastic gradient $g_t(x)$ (as an estimator of the true gradient $\nabla f_t(x)$) satisfies that for all $x \in \mathbb{R}^d$, $\mathbb{E}\left[g_t(x)\right] = \nabla f_t(x)$ and $\mathbb{E}\left[\|g_t(x) - \nabla f_t(x)\|^2\right] \leq \sigma_g^2$, where $\sigma_g^2$ is the variance of the stochastic gradient feedback.*

### 4.1 LOWER BOUND ON LOCAL REGRET WITH SSGO

We first establish a novel lower bound on the local regret of the class of linear-span algorithms defined in Definition 1 in the nonconvex SSGO setting.

**Theorem 3.** *Suppose that an algorithm $\mathcal{A}$ satisfies Definition 1 with $\nabla f_t(x_t)$ replaced by its stochastic version $g_t(x_t)$. Then, for problems with the dimension $d \geq \Omega\left(\sqrt{T(1 + V_T)}\right)$, there exist $\{f_t\}_{t=1}^{\infty}$ with the function variation $V_T$ (defined in eq. (2)) and satisfying Assumption 1 as well as $\{g_t\}_{t=1}^{\infty}$ satisfying Assumption 2, for which $\mathbb{E}\left[\mathfrak{R}(T)\right] \geq \Omega\left(\sigma_g\sqrt{T(1 + V_T)}\right)$.*

*Proof Outline of Theorem 3.* The detailed proof is provided in Appendix B.2. Rather than taking the sigmoid function as the basic example function in the proof of Theorem 1, we adopt the components of hard-to-learn function in offline nonconvex optimization (Carmon et al., 2021; Arjevani et al., 2022) to construct the hard case here, to capture how the randomness of the stochastic gradient can hinder the learning process. □

Theorem 3 provides the first meaningful lower bound on the local regret for online nonconvex optimization with SSGO feedback. It captures how the regret scales with the variance of stochastic gradients, the number of iterations, and the function variation. The existing result only provides a vacuous $\Omega(T)$ lower bound by taking $w = 1$ in Guan et al. (2023, Theorem 1).

Compared to the lower bound in Theorem 1 for the SGO setting, the lower bound in Theorem 3 is larger due to: (a) $\sqrt{T(1 + V_T)} \geq \Omega(V_T)$ because $V_T \leq \mathcal{O}(T)$ and (b) the positive variance $\sigma_g$. Hence, stochastic gradients cause the problem to be harder than exact gradients.

### 4.2 LOCAL REGRET OF ONLINE STOCHASTIC GRADIENT DESCENT

In this subsection, we show that the standard online stochastic gradient descent (OSGD) algorithm achieves the lower bound on the local regret in the nonconvex setting with SSGO feedback.

---

**Algorithm 2** Online Stochastic Gradient Descent (OSGD)

---

**Input:** Initial point $x_1$, stepsizes $\eta$
**for** $t = 1, \ldots, T$ **do**
    Update $x_{t+1}$ based on eq. (4)
**end for**

---

At each $t$, the OSGD algorithm (see Algorithm 2) with constant stepsize $\eta > 0$ takes the update

$$x_{t+1} = x_t - \eta g_t(x_t), \tag{4}$$

where $g_t$ is the stochastic gradient that satisfies Assumption 2.

Our lower bound in Theorem 3 implies that the lowest possible regret for the class of linear-span algorithms to expect is $\mathcal{O}\left(\sigma_g \sqrt{T(1 + V_T)}\right)$. The following theorem shows that such an optimal rate for linear-span algorithms is attained by OSGD.

**Theorem 4.** *Suppose Assumptions 1 and 2 hold. Consider Algorithm 2 with initial point $x_1 = 0$, and stepsize $\eta = \min\left\{\frac{1}{2L_1}, \frac{1}{\sigma_g}\sqrt{\frac{1+V_T}{T}}\right\}$. Then, we have $\mathbb{E}\left[\mathfrak{R}(T)\right] \leq \mathcal{O}\left(\sigma_g\sqrt{(1 + V_T)T}\right)$.*

*Proof Outline of Theorem 4.* The detailed proof is provided in Appendix B.3. The main step lies in decomposing the regret into the tracking error of the stationary points, which is bounded by $\mathcal{O}(\frac{1+V_T}{\eta})$, and the variance of the stochastic gradient (by taking the conditional expectation given the history information) which is bounded by $\mathcal{O}(\eta\sigma_g^2)$. Then, the final regret bound can be obtained by the best tradeoff between the tracking error and the variance via the stepsize $\eta$. □

Theorem 4 provides an upper bound on the expected value of the local regret. Next, we establish a more refined characterization of the local regret with a high probability guarantee. To this end, we first state the following boundedness assumption on the stochastic gradient.

**Assumption 3.** *For any $t$ and $x \in \mathbb{R}^d$, there exists a constant $G > 0$, such that the stochastic gradient satisfies $\|g_t(x)\| \leq G$ almost surely.*

**Theorem 5.** *Suppose Assumptions 1 to 3 hold. Consider the OSGD Algorithm 2 with initial point $x_1 = 0$ and stepsize $\eta = \min\left\{\frac{1}{2L_1}, \frac{1}{\sigma_g}\sqrt{\frac{1+V_T}{T}}\right\}$. Then, for any $0 < \xi < 1$, with probability at least $1 - \xi$ we have $\mathfrak{R}(T) \leq \mathcal{O}\left(\sigma_g\sqrt{(1 + V_T)T}\right) + \mathcal{O}\left(\sigma_g\sqrt{T\log(1/\xi)}\right)$.*

Theorem 5 shows that if the stochastic gradient feedback has a bounded norm, the local regret concentrates around its expected value with high probability.

*Proof Outline of Theorem 5.* The detailed proof is provided in Appendix B.4. Below we summarize our main proof steps. In the proof of Theorem 4, we have shown that conditioned on the filtration $\mathcal{F}_t = \sigma(g_1(x_1), \ldots, g_{t-1}(x_{t-1}))$, $\|\nabla f_t(x_t)\|^2 \leq \frac{f_t(x_t) - \mathbb{E}[f_t(x_{t+1})|\mathcal{F}_t]}{(1 - L_1\eta)\eta} + \frac{L_1\eta\sigma_g^2}{2(1 - L_1\eta)}$. Based on the above property, we can construct the following super-martingale sequence $Z_0 = 0$, and $Z_{t+1} = Z_t + \|\nabla f_t(x_t)\|^2 - \frac{f_t(x_t) - f_t(x_{t+1})}{(1 - L_1\eta)\eta} - \frac{L_1\eta\sigma_g^2}{2(1 - L_1\eta)}$. It can be shown that $|Z_t - Z_{t-1}|$ is bounded by a certain constant. Then by applying the Azuma-Hoeffding inequality to $\{Z_t\}$, we can obtain a high probability upper bound on the local regret. □

## 5 ONLINE NONCONVEX OPTIMIZATION WITH SVO

In this section, we study the online nonconvex optimization problem with the agent having access to only a single function value oracle (SVO) feedback at each time $t$.

## 5.1 Algorithm Design

The setting with SVO feedback is much more challenging than the gradient feedback because with only a single function value, the gradient of the function cannot be estimated accurately. The conventional one-point gradient estimator (Flaxman et al., 2005) has been used in online optimization and is also widely adopted in offline gradient-free algorithms. The performance of such an estimator is much worse than the standard two-point gradient estimator used previously in online optimization (Agarwal et al., 2010). Albeit the great performance gain, the two-point estimator cannot be used in our setting of SVO because it requires two function value oracles at each time step.

Hence, our goal here is to design an online algorithm that requires only a single function value oracle, but achieves the same regret as that of a two-point estimator. To this end, we design the following one-point gradient estimator based on the running difference of online function values:

$$\widehat{\nabla} f_t(x_t) := \tfrac{du_t}{\delta} \left( f_t(x_t + \delta u_t) - f_{t-1}(x_{t-1} + \delta u_{t-1}) \right), \tag{5}$$

where $u_t$ is drawn from the uniform distribution over unit sphere on $\mathbb{R}^d$ independently from history. Such an estimator uses the consecutive value feedback to construct a similar residual structure as the two-point estimator. Our idea is inspired by the one-point residual estimator proposed in Zhang et al. (2022) for offline optimization but has a subtle yet critical difference. The one-point residual estimator in Zhang et al. (2022) is given by $\widehat{\nabla} f_t(x_t) := \tfrac{du_t}{\delta} \left( f_{\xi_t}(x_t + \delta u_t) - f_{\xi_{t-1}}(x_{t-1} + \delta u_{t-1}) \right)$, where $\xi_t$ and $\xi_{t-1}$ are two samples following the same distribution as the sampling variable $\xi$. Thus, both $f_{\xi_t}$ and $f_{\xi_{t-1}}$ are unbiased noisy samples of $\mathbb{E}_\xi[f_\xi]$. However, in our one-point running difference estimator, $f_t$ and $f_{t-1}$ are two online functions and can be very different. Their relationship is only captured by the function variation over time. This difference causes a significant difference in the theoretical analysis: our analysis relies on the function variation $V_T$ to bound the regret in online optimization, whereas the analysis in Zhang et al. (2022) exploits the stochastic unbiased property of the two functions in offline optimization. Compared to the one-point estimator used in Guan et al. (2023), our estimator does not skip updates in every other iteration and thus is much more sample efficient. We then propose an OGD-type algorithm (see Algorithm 3) for online nonconvex optimization with SVO feedback, which uses the one-point running difference gradient estimator given in eq. (5) to update the variable.

---

**Algorithm 3** OGD with One-point Running Difference Estimation

---

**Input:** Initial point $x_1$, stepsizes $\eta$ and parameter $\delta$
**for** $t = 1, \ldots, T$ **do**
    Draw $u_t$ from the uniform distribution over the unit sphere and observe $f_t(x_t + \delta u_t)$
    Let $\widehat{\nabla} f_t(x_t) = \tfrac{d}{\delta} \left( f_t(x_t + \delta u_t) - f_{t-1}(x_{t-1} + \delta u_{t-1}) \right) u_t$
    Let $x_{t+1} = x_t - \eta \widehat{\nabla} f_t(x_t)$
**end for**

---

## 5.2 Regret Analysis

As aforementioned, the technical analysis of Algorithm 3 in online optimization is very different from the one-point gradient residual estimator in offline optimization. In particular, our main technical development lies in providing a novel upper bound on the magnitude of the one-point running difference gradient estimator regarding the function variation (see Lemma 1). With such a property, the local regret can be further bounded via the function variation.

**Lemma 1.** *Suppose $\eta$ and $\delta$ are chosen satisfying $\eta \leq \delta/(4L_0 d)$. For Algorithm 3, we have*

$$\left\| \widehat{\nabla} f_t(x_t) \right\|^2 \leq \tfrac{1}{2^{t-1}} \left\| \widehat{\nabla} f_1(x_1) \right\|^2 + \tfrac{4d^2}{\delta^2} \sum_{\tau=2}^{t} \tfrac{(f_\tau(x_\tau) - f_{\tau-1}(x_\tau))^2}{2^{t-\tau}} + 16 d^2 L_0^2. \tag{6}$$

*Proof Outline of Lemma 1.* The proof of Lemma 1 is provided in Appendix C.1. It mainly relies on two developments: (a) extracting the function variation from the form of the gradient estimator; and (b) constructing the contraction of gradient norms by selecting sufficiently small stepsize. □

Applying Lemma 1, we establish the following upper bound on the regret for Algorithm 3.

**Theorem 6.** *Suppose Assumption 1 holds. Consider Algorithm 3 with initial point $x_1 = 0$, perturbation $\delta = \sqrt{(1 + V_T)/T}$ and stepsize $\eta = \delta/(4L_0 d)$. Then, the expected local regret satisfies $\mathbb{E}\left[\mathfrak{R}(T)\right] \leq \mathcal{O}(d\sqrt{(1 + V_T)T})$.*

The result in Theorem 6 indicates that our one-point running difference gradient estimator (when applied to OGD) achieves the regret lower bound that a generic stochastic gradient oracle can best achieve (i.e., Theorem 3), where the variance of our gradient estimator is at the order of $\mathcal{O}(d^2)$. We will also show below that our one-point estimator yields the same regret as the standard two-point estimator. Both of the above facts indicate that Algorithm 3 achieves a desirable regret performance.

*Proof Outline of Theorem 6.* The detailed proof is provided in Appendix C.2. Differently from the proof of Theorem 4, the main technical development here lies in upper-bounding the bias and variance of our one-point running difference gradient estimator. More specially, the bias term satisfies $\mathbb{E}\left[\widehat{\nabla}f_t(x_t)|\mathcal{F}_t\right] = \mathbb{E}\left[\frac{d}{\delta}f_t(x_t + \delta u_t)u_t|\mathcal{F}_t\right] = \nabla f_{t,\delta}(x_t)$, which can be shown to be bounded by $\mathcal{O}(\delta)$. The variance term has been bounded in Lemma 1. Then, the tradeoff between the bias, the variance, and the tracking error of the stationary points via $\delta$ and $\eta$ provides the best regret bound. $\square$

**Comparison to standard one-point and two-point estimators:** Previous online optimization studies (Flaxman et al., 2005) mainly adopted the conventional one-point gradient estimator, for which it can be shown (see Appendix D.1) that the local regret of OSGD with such an estimator in online nonconvex optimization is bounded as $\mathbb{E}\left[\mathfrak{R}(T)\right] \leq \mathcal{O}(d(1 + V_T)^{\frac{1}{4}}T^{\frac{3}{4}})$. As a comparison, Theorem 6 obtained by our one-point running difference estimator achieves much smaller regret (note that $V_T$ scales at most with the order of $T$, and can scale much slower than $T$). Furthermore, it can be shown that the local regret of OSGD with standard two-point gradient estimator (Agarwal et al., 2010) satisfies $\mathbb{E}\left[\mathfrak{R}(T)\right] \leq \mathcal{O}(d\sqrt{(1 + V_T)T})$ (in Appendix D.2). Clearly, Theorem 6 shows that our one-point estimator achieves the same regret as the two-point estimator. This is the first result in online nonconvex optimization that establishes that one function value oracle achieves the same regret as two oracles.

Since Lemma 1 has already provided an upper bound on $\|\widehat{\nabla}f_t(x_t)\|$, we can further obtain the following high probability upper bound on the local regret without making a further assumption on the norm of "gradient" as in Assumption 3.

**Theorem 7.** *Suppose Assumption 1 holds. Consider Algorithm 3 with initial point $x_1 = 0$, perturbation $\delta = (1 + V_T)/T$, and stepsize $\eta = \delta/(4L_0 d)$. Then, for any $0 < \xi < 1$, with probability at least $1 - \xi$ we have $\mathfrak{R}(T) \leq \mathcal{O}\left(d\sqrt{T(1 + V_T)\log(1/\xi)}\right)$.*

*Proof Outline of Theorem 7.* The detailed proof is provided in Appendix C.3. Below we summarize our main idea of the proof. In the proof of Theorem 6, we have obtained the following bound:
$$\|\nabla f_t(x_t)\| \leq \frac{f_t(x_t) - f_t(x_{t+1})}{\eta} + L_1 L_0 d\delta + \frac{L_1 \eta}{2}\mathbb{E}\left[\|\widehat{\nabla}f_t(x_t)\|^2 \Big| \mathcal{F}_t\right].$$

Based on the above bound, we construct the following super-martingale: $Y_0 = 0$, and $Y_{t+1} = Y_t + \|\nabla f_t(x_t)\|^2 + \frac{f_t(x_{t+1}) - f_t(x_t)}{\eta} - L_1 L_0 d\delta - \frac{L_1 \eta}{2}\|\widehat{\nabla}f_t(x_t)\|^2$. We then construct a sequence of scalar $c_t$ such that $c_t \geq |Y_{t+1} - Y_t|$. Finally, we obtain the probability bound on the local regret by applying Azuma-Hoeffding inequality and upper-bounding $\sum_{t=1}^{T} c_t^2$. $\square$

## 6 CONCLUSION

In this paper, we investigate the online nonconvex optimization with a single oracle feedback per time step. We take the local regret of original objective functions as performance metric and study three single oracle settings: with an (exact) gradient oracle, with a stochastic gradient oracle, and with a function value oracle. We provide the first regret lower bound for both the exact and stochastic gradient oracles, and we show that the online (stochastic) gradient descent can achieve the optimal local regret for the class of linear-span algorithms. For the setting with function value oracle, we propose one-point running difference gradient estimator and show that incorporating such an estimator into online gradient descent achieves a local regret that a generic stochastic gradient oracle can best achieve for the class of linear-span algorithms.

ACKNOWLEDGMENTS

The work of Z. Guan and Y. Liang was supported in part by the U.S. National Science Foundation under the grants CCF-1900145 and CCF-1761506.

Y. Zhou's work is supported by the National Science Foundation under grants CCF-2106216, DMS-2134223, ECCS-2237830 (CAREER).

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

## Supplementary Materials

## A   PROOFS OF SECTION 3

In this section, we provide the proofs for Theorems 1 and 2 in Section 3.

### A.1   PROOF OF THEOREM 1

The main idea of the proof is as follows. Given a total budget of $V_T$ on the function variation, the objective function can have $\Omega(1 + V_T)$ rapid changes. Thus, we divide the total time steps into $\Omega(1 + V_T)$ blocks, choose the same objective function within each block and change the objective function across blocks. We then construct a series of functions whose gradients are orthogonal to each other and assign them to these blocks, which hinders the agent from constructing $\nabla f_t(x_t)$ based on feedback from previous blocks. This construction of $\{f_t\}_{t=1}^T$ forces the agent to restart the learning process in each block. The agent suffers from a high value of $\|\nabla f_t(x_t)\|^2$ at the beginning of each restart, and thus the local regret is doomed to be $\Omega(1 + V_T)$.

To present the detailed proof, we first specify the objective functions for each block. Define the one-entry sigmoid functions as

$$\tilde{f}_k(x) = \frac{c \exp([x]_k)}{1 + \exp([x]_k)}, \tag{7}$$

where $c > 0$ is a constant that will be determined later, and $[x]_k$ is the $k$-th entry of the vector $x$. Clearly, we have, $0 \le \tilde{f}_k(x) \le c$, and

$$\nabla \tilde{f}_k(x) = \frac{c \exp([x]_i)}{(1 + \exp([x]_i))^2} e_k,$$

with $e_k$ is the $k$th basis of the Euclidean space. Then we have, for every $x$ and $y$,

$$|\tilde{f}_k(x) - \tilde{f}_k(x)| \le \sup_{x \in \mathbb{R}^d} |\nabla \tilde{f}_k(x)| \le \frac{c}{2}$$

and

$$\|\nabla \tilde{f}_k(x) - \nabla \tilde{f}_k(y)\| \le \frac{\sqrt{3}c}{18} \|x - y\|.$$

Therefore, by taking $c = \min\{M, 2L_0, 6\sqrt{3}L_1\}$, each $\tilde{f}_k(x)$ satisfies Assumption 1.

We now divide the total $T$ iterations into $\left\lceil \frac{1+V_T}{c} \right\rceil - 2$ blocks, and let $B = \left\lfloor \frac{cT}{V_T+1} \right\rfloor$ be the length of each block. Within each block, we fix the objective function, i.e., for all $k$,

$$f_{(k-1)B+1}(x) = f_{(k-1)B+2}(x) = \ldots = f_{kB}(x) = \tilde{f}_k(x).$$

Thus, due to our construction, we have

$$V_T^f := \sum_{t=2}^{T+1} \sup_{x \in \mathbb{R}^d} |f_{t-1}(x) - f_t(x)| = \sum_{k=1}^{\left\lceil \frac{1+V_T}{c} \right\rceil - 2} \sup_{x \in \mathbb{R}^d} |f_{kK+1}(x) - f_{kK}(x)|$$

$$\le c \left( \left\lceil \frac{1 + V_T}{c} \right\rceil - 2 \right) \le V_T.$$

Due to our construction, for the $k$-th block, the gradients received from the previous $k - 1$ block reveal only the first $k - 1$ dimensions. Thus, by Definition 1, $[x_{kB+1}]_k = 0$ for all $k$. And we have

$$\|\nabla f_{kB+1}(x_{kB+1})\|^2 = \frac{c^2}{4}.$$

Therefore, we have

$$\mathfrak{R}(T) = \sum_{t=1}^T \|\nabla f_t(x_t)\|^2 \ge \sum_{k=1}^{\left\lceil \frac{1+V_T}{c} \right\rceil - 2} \|\nabla f_{kB+1}(x)\|^2 = c^2 \left( \left\lceil \frac{1 + V_T}{c} \right\rceil - 2 \right) = \Omega(1 + V_T).$$

## A.2 Proof of Theorem 2

The main idea for this proof is to leverage the Lipschitz smoothness of each objective function to connect the local regret to the cumulative difference $\sum_{t=1}^{T} f_{t+1}(x_{t+1}) - f_t(x_{t+1})$ that is upper-bounded by $V_T$. By the $L_1$-gradient Lipshitz condition of $f_t(x)$, we have

$$
\begin{aligned}
f_t(x_{t+1}) &\leq f_t(x_t) + \langle \nabla f_t(x_t), x_{t+1} - x_t \rangle + \frac{L_1}{2} \|x_{t+1} - x_t\|^2 \\
&\overset{(i)}{=} f_t(x_t) - \left( \eta - \frac{L_1 \eta^2}{2} \right) \|\nabla f_t(x_t)\|^2 \\
&\overset{(ii)}{=} f_t(x_t) - \frac{1}{2L_1} \|\nabla f_t(x_t)\|^2,
\end{aligned}
\tag{8}
$$

where $(i)$ follows from the update rule in eq. (3) and $(ii)$ follows because $\eta = \frac{1}{L_1}$.

Rearranging eq. (8) and telescoping, we have

$$
\begin{aligned}
\mathfrak{R}(T) &= \sum_{t=1}^{T} \|\nabla f_t(x_t)\|^2 \\
&\leq 2L_1 \sum_{t=1}^{T} (f_t(x_t) - f_t(x_{t+1})) \\
&= 2L_1(f_1(x_1) - f_{T+1}(x_{T+1})) + 2L_1 \sum_{t=1}^{T} (f_{t+1}(x_{t+1}) - f_t(x_{t+1})) \\
&\overset{(i)}{\leq} 4L_1 M + 2L_1 V_T,
\end{aligned}
$$

where $(i)$ follows from the upper bound of the function value and the definition of $V_T$.

## B Proofs of Section 4

In this section, we first introduce the hard-to-learn nonconvex function constructed by Carmon et al. (2020). This function is utilized next in the proof of Theorem 3. We then provide the proofs of Theorems 3 to 5.

### B.1 Hard-to-learn Nonconvex function

In Carmon et al. (2020), the authors constructed a special nonconvex function that is hard for a gradient-based algorithm to learn. The function $F_K : \mathbb{R}^K \to \mathbb{R}$ is defined as

$$
F_K(x) = -\Psi(1)\Phi([x]_1) + \sum_{i=2}^{K} [\Psi(-[x]_{i-1})\Phi(-[x]_i) - \Psi([x]_{i-1})\Phi([x]_i)],
\tag{9}
$$

where $K$ is a certain integer, $[x]_i$ denotes the $i$th entry of $x$, and the $\Psi(z)$ and $\Phi(z)$ are defined as

$$
\Psi(z) := \begin{cases} 0 & z \leq 1/2 \\ \exp\left(1 - \frac{1}{(2z-1)^2}\right) & z > 1/2 \end{cases},
$$

and

$$
\Phi(z) := \sqrt{e} \int_{-\infty}^{z} \exp\left(-\frac{t^2}{2}\right).
$$

The hard function satisfies the following properties (Carmon et al., 2020).

$$
|F_K(x)| \leq 24K \tag{10}
$$

$$
\|\nabla F_K(x)\| \leq 23\sqrt{K} \tag{11}
$$

$$
\|\nabla F_K(x) - \nabla F_K(y)\| \leq 152\|x - y\|. \tag{12}
$$

### B.2 PROOF OF THEOREM 3

Rather than taking the sigmoid function as the basic example function in the proof of Theorem 1, we adopt the components of the aforementioned hard-to-learn function in eq. (9) to construct the hard case here, to capture how the randomness of the stochastic gradient can hinder the learning process.

We divide the iterations into blocks with size $B$, and we set $B = \left\lceil \frac{2M(T+1)}{1+V_T} \right\rceil$. Denote $\mathcal{T}_k$ to be the $k$-th block containing the iterations $(k-1)B+1, (k-1)B+1, \ldots, kB$. Moreover, the objective functions $f_t$ within $\mathcal{T}_{k+1}$ are fixed to be a same function $\hat{f}_k(x)$, i.e., for all $k$,

$$f_{kB+1}(x) = f_{kB+2}(x) = \cdots = f_{(k+1)B}(x) = \hat{f}_k(x),$$

where $\hat{f}_k(x)$ is defined as

$$
\begin{aligned}
\hat{f}_k&(x) \\
&= c\lambda^2 F_K \left( \frac{[x]_{\{(k-1)B+1,\ldots,kB\}}}{\lambda} \right) \\
&= -c\lambda^2 \Psi(1)\Phi([x]_{(k-1)B+1}) \\
&\quad + c\lambda^2 \sum_{i=2}^{K} [\Psi(-[x]_{(k-1)B+i-1})\Phi(-[x]_{(k-1)B+i}) - \Psi([x]_{(k-1)B+i-1})\Phi([x]_{(k-1)B+i})],
\end{aligned}
$$
(13)

where $c = \frac{L_1}{152}$, $\lambda = \frac{1}{\sqrt{K}} \min\left\{ \frac{152L_0}{23L_1}, \sqrt{\frac{19M}{3L_1}} \right\}$, and $K$ is an integer satisfying $\lceil \sigma_g^2 K^2 \rceil = B$. Moreover, $[x]_i$ denotes the $i$-th entry of $x$, and $[x]_{\{a,\ldots,b\}}$ denotes the $(b-a+1)$-dimensional vector consisting of the entries $[x]_a, \ldots, [x]_b$.

We now verify that $\hat{f}_k(x)$ satisfies Assumption 1. Using eq. (10), we obtain

$$\left| c\lambda^2 F_K \left( \frac{[x]_{\{(k-1)B+1,\ldots,kB\}}}{\lambda} \right) \right| \leq \frac{2}{4} c\lambda^2 K \leq M.$$
(14)

Next, by eq. (11), we have

$$
\begin{aligned}
\left\| \nabla \left( c\lambda^2 F_K \left( \frac{[x]_{\{(k-1)B+1,\ldots,kB\}}}{\lambda} \right) \right) \right\| &= c\lambda \left\| \nabla F_K \left( \frac{[x]_{\{(k-1)B+1,\ldots,kB\}}}{\lambda} \right) \right\| \\
&\leq 23c\sqrt{K}\lambda \leq L_0,
\end{aligned}
$$
(15)

which implies the $L_0$-Lipschitz continuous property.

Moreover, by eq. (12), we obtain

$$
\begin{aligned}
\left\| \nabla \left( c\lambda^2 F_K \left( \frac{[x]_{\{(k-1)B+1,\ldots,kB\}}}{\lambda} \right) \right) - \nabla \left( c\lambda^2 F_K \left( \frac{[y]_{\{(k-1)B+1,\ldots,kB\}}}{\lambda} \right) \right) \right\| & \\
= c\lambda \left\| \nabla F_K \left( \frac{[x]_{\{(k-1)B+1,\ldots,kB\}}}{\lambda} \right) - \nabla F_K \left( \frac{[y]_{\{(k-1)B+1,\ldots,kB\}}}{\lambda} \right) \right\| & \\
\leq 152c\|x-y\| & \\
\leq L_1\|x-y\|. &
\end{aligned}
$$
(16)

The eqs. (14) to (16) prove that every $f_t$ satisfies Assumption 1. We next show that our choice of the length $B$ of each block ensures that the function variation does not exceed $V_T$. The function variation $V_T^f$ of our construction is upper-bounded as follows:

$$V_T^f := \sum_{t=2}^{T+1} \sup_{x \in \mathbb{R}^d} |f_{t-1}(x) - f_t(x)| = \sum_{k=1}^{\lfloor \frac{T+1}{B} \rfloor} \sup_{x \in \mathbb{R}^d} |f_{kK+1}(x) - f_{kK}(x)| \overset{(i)}{\leq} 2M \left\lfloor \frac{T+1}{B} \right\rfloor,$$

where $(i)$ follows from eq. (14). Since we set $B = \left\lceil \frac{2M(T+1)}{1+V_T} \right\rceil$, we have $V_T^f \leq V_T$. And, the choice of $K$ is determined immediately by the relationship $\lceil \sigma_g^2 K^2 \rceil = B$.

Lastly, we let the stochastic gradient oracle $g_t(x)$ be

$$[g_t(x)]_i = [\nabla f_t(x)]_i \cdot \left(1 + \mathbb{1}\left\{i > \text{proj}_{\frac{1}{4}}(x)\right\}\left(\frac{z_t}{p} - 1\right)\right),\qquad(17)$$

where $\text{proj}_{\frac{1}{4}}(x) := \max\left\{i \geq 0\big|\,|[x]_i| \geq \frac{1}{4}\right\}$, and $z_t \sim \text{Bernoulli}(p)$ with $p = \min\left\{1, \frac{23^2 c^2 \lambda^2}{\sigma_g^2}\right\}$. Clearly $g_t(x)$ is unbiased and, by Arjevani et al. (2022, Lemma 3), the variance of $g_t(x)$ is smaller than $\sigma_g$. Hence, the construction of $g_t(x)$ here satisfies Assumption 2.

Based on our designed functions $f_t$ and the stochastic gradient oracles $g_t$, we next prove the regret lower bound. At each block $k$, the stochastic gradient oracle feedback obtained from the previous $k-1$ blocks all lie in the first $(k-1)K$ dimensions of $x$. Therefore, due to Definition 1, we have $[x_{(k-1)K}]_{(k-1)K+1,\ldots,kK} = \mathbf{0}$. And the objective function within the $k$-th block depends only on the $(k-1)K + 1$-th of $x$ up to its $kK$-th entry. Thus, the optimization within the $k$-th block equals the offline optimization with total $K$ iterations of updates. Applying Arjevani et al. (2022, Lemma 1), with probability greater or equal to $\frac{1}{2}$, for all $kB + 1 \leq t \leq kB + \min\left\{\frac{K - \log(2)}{2p}, B\right\}$, we have

$$\|\nabla f_t(x_t)\| \geq c\lambda = \Omega\left(\frac{1}{\sqrt{K}}\right).\qquad(18)$$

We then conclude from eq. (18) that

$$\mathbb{E}\left[\sum_{i=1}^{B}\|\nabla f_{kB+i}(x_{kB+i})\|^2\right] \geq \Omega\left(\frac{1}{K}\right)\cdot\min\left\{\frac{K - \log(2)}{2p}, B\right\} = \Omega\left(\sigma_g^2 K\right).\qquad(19)$$

We next derive the following bound on the expected value of the local regret:

$$\begin{aligned}
\mathbb{E}\left[\mathfrak{R}(T)\right] &= \mathbb{E}\left[\sum_{t=1}^{T}\|\nabla f_t(x)\|^2\right]\\
&= \mathbb{E}\left[\sum_{k=1}^{\lfloor\frac{T+1}{B}\rfloor+1}\sum_{i=1}^{K}\|\nabla f_{kB+i}(x)\|^2\right]\\
&\overset{(i)}{\geq} \Omega(\sigma_g^2 K)\cdot\left\lfloor\frac{T+1}{B}\right\rfloor\\
&= \Omega(\sigma_g\sqrt{T(1+V_T)}).
\end{aligned}\qquad(20)$$

where $(i)$ follows from eq. (19).

### B.3 Proof of Theorem 4

The main step lies in decomposing the regret into the tracking error of the stationary points which is bounded by $\mathcal{O}\left(\frac{1+V_T}{\eta}\right)$, and the variance of the stochastic gradient (by taking the conditional expectation given the history information) which is bounded by $\mathcal{O}\left(\eta\sigma_g^2\right)$. Then, the final regret bound can be obtained by the best tradeoff between the tracking error and the variance via the stepsize $\eta$.

To proceed the proof, following from the $L_1$-gradient Lipshitz condition of $f_t(x)$, we have

$$\begin{aligned}
f_t(x_{t+1}) &\leq f_t(x_t) + \langle\nabla f_t(x_t), x_{t+1} - x_t\rangle + \frac{L_1}{2}\|x_{t+1} - x_t\|^2\\
&\overset{(i)}{=} f_t(x_t) - \eta\langle\nabla f_t(x_t), g_t(x_t)\rangle + \frac{L_1\eta^2}{2}\|g_t(x_t)\|^2\\
&\overset{(ii)}{\leq} f_t(x_t) - \eta\langle\nabla f_t(x_t), g_t(x_t)\rangle + L_1\eta^2\|g_t(x_t) - \nabla f_t(x_t)\|^2 + L_1\eta^2\|\nabla f_t(x_t)\|^2,\quad(21)
\end{aligned}$$

where $(i)$ follows from the update rule in eq. (4) and $(ii)$ follows from the Young's inequality.

Taking expectation conditioned on $x_t$ on both sides of eq. (21) and applying Assumption 2, we have

$$\mathbb{E}\left[f_t(x_{t+1})|x_t\right] \leq f_t(x_t) - \eta\left(1 - L_1\eta\right)\|\nabla f_t(x_t)\|^2 + L_1\eta^2\sigma_g^2.\qquad(22)$$

Taking expectation on eq. (22), rearranging and telescoping, we have

$$\mathbb{E}\left[\mathfrak{R}(T)\right]$$

$$= \sum_{t=1}^{T} \mathbb{E}\left[\|\nabla f_t(x_t)\|^2\right]$$

$$\leq \frac{1}{\eta(1-L_1\eta)} \sum_{t=1}^{T} \mathbb{E}\left[f_t(x_t) - f_t(x_{t+1})\right] + \frac{\eta\sigma_g^2 T}{1-L_1\eta}$$

$$= \frac{1}{\eta(1-L_1\eta)} \left( \mathbb{E}\left[f_1(x_1) - f_{T+1}(x_{T+1})\right] + \sum_{t=1}^{T} \mathbb{E}\left[f_{t+1}(x_{t+1}) - f_t(x_{t+1})\right] \right) + \frac{\eta\sigma_g^2 T}{1-L_1\eta}$$

$$\overset{(i)}{\leq} \frac{1}{\eta(1-L_1\eta)}(2M + 2V_T) + \frac{\eta\sigma_g^2 T}{1-L_1\eta}, \tag{23}$$

where $(i)$ follows from the upper bound of the function value and the definition of $V_T$. Applying the definition of $\eta$, we complete the proof.

### B.4 Proof of Theorem 5

Define a sequence of random variable $\{Z_t\}_{t=1}^{\infty}$ as

$$Z_0 = 0 \quad \text{and} \quad Z_{t+1} = Z_t + \frac{f_t(x_{t+1}) - f_t(x_t)}{\eta} + (1-L_1\eta)\|\nabla f_t(x_t)\|^2 - L_1\eta\sigma_g^2.$$

Define $\mathcal{F}_t = \sigma(x_1, x_2, \dots, x_t)$. Due to eq. (22), we have

$$\mathbb{E}\left[Z_{t+1}|\mathcal{F}_t\right] \leq Z_t.$$

Moreover,

$$|Z_{t+1} - Z_t| = \left| \frac{f_t(x_{t+1}) - f_t(x_t)}{\eta} + (1-L_1\eta)\|\nabla f_t(x_t)\|^2 - \eta\sigma_g^2 \right|$$

$$\overset{(i)}{\leq} L_0 G + L_1^2 + \sigma_g L_1 M,$$

where $(i)$ follows because

$$|f_t(x_{t+1}) - f_t(x_t)| \leq L_0\|x_t - x_{t+1}\| = L_0\eta\|g_t(x_t)\| \leq L_0 G\eta,$$

and $\|\nabla f_t(x_t)\|^2 \leq L_1^2$, and $\eta \leq \frac{M}{\sigma_g}$.

Therefore, $\{Z_t\}_{t=1}^{\infty}$ is a super-martingale with a bounded difference. Applying the Azuma–Hoeffding inequality to $Z_{T+1}$, we have

$$\Pr\left(Z_{T+1} - Z_0 \geq \epsilon\right) \leq 2\exp\left(-\frac{\epsilon^2}{2T(L_0 G + L_1^2 + \sigma_g L_1 M)^2}\right).$$

By taking $\epsilon = \sqrt{2T(L_0 G + L_1^2 + \sigma_g L_1 M)^2 \log\left(\frac{2}{\xi}\right)}$, we have, with probability greater or equal to $1-\xi$,

$$Z_{T+1} \leq \sqrt{2T(L_0 G + L_1^2 + \sigma_g L_1 M)^2 \log\left(\frac{2}{\xi}\right)}.$$

Using the definition of $Z_{T+1}$ and rearranging the terms, we have, with probability great or equal to $1-\xi$,

$$\mathfrak{R}(T) = \sum_{t=1}^{T} \|\nabla f_t(x_t)\|^2$$

$$\leq \frac{1}{\eta(1-L_1\eta)}(2M + 2V_T) + \frac{1}{1-L_1\eta}\frac{\eta\sigma_g^2 T}{1-L_1\eta} + \sqrt{2T(L_0 G + L_1^2 + \sigma_g L_1 M)^2 \log\left(\frac{2}{\xi}\right)}.$$

Substituting the definition of $\eta$ into the above inequality completes the proof.

## C  PROOFS OF SECTION 5

### C.1  PROOF OF LEMMA 1

We upper-bound the variance term $\|\widehat{\nabla} f_t(x_t)\|^2$ defined in eq. (5) as follows.

$$
\left\| \widehat{\nabla} f_t(x_t) \right\|^2
$$

$$
= \frac{d^2}{\delta^2} \left( f_t(x_t + \delta u_t) - f_{t-1}(x_{t-1} + \delta u_{t-1}) \right)^2
$$

$$
= \frac{d^2}{\delta^2} \big( f_t(x_t + \delta u_t) - f_t(x_t) + f_t(x_t) - f_{t-1}(x_t) + f_{t-1}(x_t) - f_{t-1}(x_{t-1})
$$

$$
+ f_{t-1}(x_{t-1}) - f_{t-1}(x_{t-1} + \delta u_{t-1}) \big)^2
$$

$$
\leq \frac{4d^2}{\delta^2} (f_t(x_t + \delta u_t) - f_t(x_t))^2 + \frac{4d^2}{\delta^2} (f_t(x_t) - f_{t-1}(x_t))^2 + \frac{4d^2}{\delta^2} (f_{t-1}(x_t) - f_{t-1}(x_{t-1}))^2
$$

$$
+ \frac{4d^2}{\delta^2} (f_{t-1}(x_{t-1}) - f_{t-1}(x_{t-1} + \delta u_{t-1}))^2
$$

$$
\overset{(i)}{\leq} \frac{4L_0^2 d^2 \eta^2}{\delta^2} \left\| \widehat{\nabla} f_{t-1}(x_{t-1}) \right\|^2 + \frac{4d^2}{\delta^2} (f_t(x_t) - f_{t-1}(x_t))^2 + 8d^2 L_0^2, \tag{24}
$$

where $(i)$ follows because $f_t$ and $f_{t-1}$ are $L_0$-Lipschitz continuous, and $x_t - x_{t-1} = \eta \widehat{\nabla} f_{t-1}(x_{t-1})$.

Since $\eta \leq \frac{\delta}{4L_0 d}$, we have $\frac{4L_0^2 d^2 \eta^2}{\delta^2} \leq \frac{1}{2}$. Iteratively applying eq. (24), we obtain

$$
\left\| \widehat{\nabla} f_t(x_t) \right\|^2 \leq \frac{1}{2^t} \left\| \widehat{\nabla} f_0(x_0) \right\|^2 + \frac{4d^2}{\delta^2} \sum_{\tau=1}^{t} \frac{(f_\tau(x_\tau) - f_{\tau-1}(x_\tau))^2}{2^{t-\tau}} + 16 d^2 L_0^2. \tag{25}
$$

### C.2  PROOF OF THEOREM 6

Differently from the proof of Theorem 4, the main technical development here lies in upper-bounding the bias and variance of our one-point running difference gradient estimator. More specially, the bias term satisfies $\mathbb{E}\left[ \widehat{\nabla} f_t(x_t) | \mathcal{F}_t \right] = \mathbb{E}\left[ \frac{d}{\delta} f_t(x_t + \delta u_t) u_t | \mathcal{F}_t \right] = \nabla f_{t,\delta}(x_t)$, which can be shown to be bounded by $\mathcal{O}(\delta)$. The variance term has been bounded in Lemma 1. Then, the tradeoff between the bias, the variance, and the tracking error of the stationary points via $\delta$ and $\eta$ provides the best regret bound.

To proceed the proof, following from the Lipschitz-smooth condition, we have

$$
f_t(x_{t+1}) \leq f_t(x_t) + \langle \nabla f_t(x_t), x_{t+1} - x_t \rangle + \frac{L_1}{2} \|x_{t+1} - x_t\|^2
$$

$$
= f_t(x_t) - \eta \langle \nabla f_t(x_t), \widehat{\nabla} f_t(x_t) \rangle + \frac{L_1 \eta^2}{2} \|\widehat{\nabla} f_t(x_t)\|^2. \tag{26}
$$

Taking expectation on both sides of eq. (26) conditioned on $\mathcal{F}_t := \sigma(u_1, \ldots, u_{t-1})$, we obtain

$$
\mathbb{E}\left[ f_t(x_{t+1}) | \mathcal{F}_t \right]
$$

$$
\leq f_t(x_t) - \eta \left\langle \nabla f_t(x_t), \mathbb{E}\left[ \widehat{\nabla} f_t(x_t) \Big| \mathcal{F}_t \right] \right\rangle + \frac{L_1 \eta^2}{2} \mathbb{E}\left[ \|\widehat{\nabla} f_t(x_t)\|^2 \Big| \mathcal{F}_t \right]
$$

$$
= f_t(x_t) - \eta \left\langle \nabla f_t(x_t), \nabla f_t(x_t) + \mathbb{E}\left[ \widehat{\nabla} f_t(x_t) \Big| \mathcal{F}_t \right] - \nabla f_t(x_t) \right\rangle + \frac{L_1 \eta^2}{2} \mathbb{E}\left[ \|\widehat{\nabla} f_t(x_t)\|^2 \Big| \mathcal{F}_t \right]
$$

$$
\overset{(i)}{\leq} f_t(x_t) - \eta \|\nabla f_t(x_t)\|^2 + \eta L_0 \left\| \mathbb{E}\left[ \widehat{\nabla} f_t(x_t) \Big| \mathcal{F}_t \right] - \nabla f_t(x_t) \right\| + \frac{L_1 \eta^2}{2} \mathbb{E}\left[ \|\widehat{\nabla} f_t(x_t)\|^2 \Big| \mathcal{F}_t \right]
$$

$$
\overset{(ii)}{\leq} f_t(x_t) - \eta \|\nabla f_t(x_t)\|^2 + \frac{L_1 L_0 d \delta \eta}{2} + \frac{L_1 \eta^2}{2} \mathbb{E}\left[ \|\widehat{\nabla} f_t(x_t)\|^2 \Big| \mathcal{F}_t \right], \tag{27}
$$

where $(i)$ follows from the Cauchy-Schwartz inequality and because $\|\nabla f_t(x)\| \leq L_0$, and $(ii)$ follows from the following Lemma 2.

**Lemma 2.** *For every t, we have*

$$\left\| \mathbb{E}\left[\widehat{\nabla} f_t(x_t) \Big| \mathcal{F}_t\right] - \nabla f_t(x_t) \right\| \leq \frac{L_1 d\delta}{2}.$$

*Proof.* Using the definition of $\widetilde{\nabla} f_t(x_t)$, we have

$$
\begin{aligned}
&\left\| \mathbb{E}\left[\widehat{\nabla} f_t(x_t) \Big| \mathcal{F}_t\right] - \nabla f_t(x_t) \right\| \\
&= \left\| \mathbb{E}\left[\frac{du_t}{\delta} f_t(x_t + \delta u_t) \Big| \mathcal{F}_t\right] - \nabla f_t(x_t) \right\| \\
&\overset{(i)}{=} \left\| \mathbb{E}\left[\frac{du_t}{2\delta}\left(f_t(x_t + \delta u_t) - f_t(x_t - \delta u_t)\right) - d\langle \nabla f_t(u_t), u_t\rangle u_t \Big| \mathcal{F}_t\right] \right\| \\
&\leq \frac{d}{\delta}\mathbb{E}\left[\left\| u_t\left(\frac{f_t(x_t + \delta u_t) - f_t(x_t - \delta u_t)}{2} - \langle \nabla f_t(x_t), \delta u_t\rangle\right)\right\|\right] \\
&\overset{(ii)}{\leq} \frac{L_1 d\delta}{2},
\end{aligned}
$$

where $(i)$ follows from Agarwal et al. (2010, Lemma 7) and $(ii)$ follows because $|f_t(x_t + \delta u_t) - f_t(x_t) + \langle \nabla f_t(x_t), \delta u_t\rangle| \leq \frac{L_1\delta^2}{2}$, $|f_t(x_t - \delta u_t) - f_t(x_t) + \langle \nabla f_t(x_t), \delta u_t\rangle| \leq \frac{L_1\delta^2}{2}$, and

$$
\begin{aligned}
&\left|\frac{f_t(x_t + \delta u_t) - f_t(x_t - \delta u_t)}{2} - \langle \nabla f_t(x_t), \delta u_t\rangle\right| \\
&= \left|\frac{f_t(x_t + \delta u_t) - f_t(x_t)\langle \nabla f_t(x_t), \delta u_t\rangle}{2} - \frac{f_t(x_t - \delta u_t) - f_t(x_t) + \langle \nabla f_t(x_t), \delta u_t\rangle}{2}\right| \\
&\leq \left|\frac{f_t(x_t + \delta u_t) - f_t(x_t) - \langle \nabla f_t(x_t), \delta u_t\rangle}{2}\right| + \left|\frac{f_t(x_t - \delta u_t) - f_t(x_t) + \langle \nabla f_t(x_t), \delta u_t\rangle}{2}\right|.
\end{aligned}
$$

$\square$

Rearranging eq. (27), we obtain

$$\|\nabla f_t(x_t)\|^2 \leq \frac{1}{\eta}\left(f_t(x_t) - \mathbb{E}\left[f_t(x_{t+1})|\mathcal{F}_t\right]\right) + \frac{L_1 L_0\delta}{2} + \frac{L_1\eta}{2}\mathbb{E}\left[\|\widehat{\nabla} f_t(x_t)\|^2\Big|\mathcal{F}_t\right]. \tag{28}$$

Taking expectation on both sides of eq. (28) and telescoping, we obtain

$$\mathbb{E}\left[\mathfrak{R}(T)\right] \leq \frac{1}{\eta}\sum_{t=1}^{T}\mathbb{E}\left[f_t(x_t) - f_t(x_{t+1})\right] + \frac{L_1 L_0\delta T}{2} + \frac{L_1\eta}{2}\mathbb{E}\left[\sum_{t=1}^{T}\|\widehat{\nabla} f_0(x_0)\|^2\right]. \tag{29}$$

To upper-bound the last term of eq. (29), we take a summation of eq. (25) over $t = 1$ to $T$ and obtain

$$
\begin{aligned}
\sum_{t=1}^{T}\left\|\widehat{\nabla} f_t(x_t)\right\|^2 &\leq \left\|\widehat{\nabla} f_0(x_0)\right\|^2\cdot\sum_{t=1}^{T}\frac{1}{2^t} + \frac{4d^2}{\delta^2}\sum_{t=1}^{T}\sum_{\tau=1}^{t}\frac{(f_\tau(x_\tau) - f_{\tau-1}(x_\tau))^2}{2^{t-\tau}} + 16d^2 L_0^2 T \\
&\leq 2\left\|\widehat{\nabla} f_0(x_0)\right\|^2 + \frac{4d^2}{\delta^2}\sum_{t=1}^{T}\left((f_t(x_t) - f_{t-1}(x_t))^2\cdot\sum_{\tau=0}^{T-t}2^\tau\right) + 16d^2 L_0^2 T \\
&\leq 2\left\|\widehat{\nabla} f_0(x_0)\right\|^2 + \frac{8d^2}{\delta^2}\sum_{t=1}^{T}(f_t(x_t) - f_{t-1}(x_t))^2 + 16d^2 L_0^2 T. \tag{30}
\end{aligned}
$$

Substituting eq. (30) into eq. (29), we obtain

$$\mathbb{E}\left[\mathfrak{R}(T)\right] \leq \frac{1}{\eta}\sum_{t=1}^{T}\mathbb{E}\left[f_t(x_t) - f_t(x_{t+1})\right] + \frac{L_1 L_0 d\delta T}{2} + L_1\eta\|\widehat{\nabla} f_0(x_0)\|^2$$

$$+ 8L_1 L_0^2 \eta d^2 T + \frac{4L_1 \eta d^2}{\delta^2} \sum_{t=1}^{T} (f_t(x_t) - f_{t-1}(x_t))^2$$

$$= \frac{f_1(x_1) - f_{T+1}(x_{T+1})}{\eta} + \frac{1}{\eta} \sum_{t=1}^{T} \mathbb{E}\left[f_{t+1}(x_{t+1}) - f_t(x_{t+1})\right] + \frac{L_1 L_0 d \delta T}{2}$$

$$+ L_1 \eta \|\widehat{\nabla} f_0(x_0)\|^2 + 8L_1 L_0^2 \eta d^2 T + \frac{4L_1 \eta d^2}{\delta^2} \sum_{t=1}^{T} (f_t(x_t) - f_{t-1}(x_t))^2$$

$$\overset{(i)}{\leq} \frac{2M}{\eta} + \frac{V_T}{\eta} + \frac{L_1 L_0 d \delta T}{2} + \frac{4L_1 d^2 M^2 \eta}{\delta^2} + 8L_1 L_0^2 \eta d^2 T + \frac{8M L_1 \eta d^2 V_T}{\delta^2}, \quad (31)$$

where $(i)$ follows because $\|\widehat{\nabla} f_0(x_0)\| \leq \frac{2Md}{\delta}$ and

$$\sum_{t=1}^{T} (f_t(x_t) - f_{t-1}(x_t))^2 \leq 2M \sum_{t=1}^{T} |f_t(x_t) - f_{t-1}(x_t)| \leq 2M V_T.$$

Substituting the definition of $\eta$ and $\delta$ into eq. (31), we obtain

$$\mathbb{E}[\mathfrak{R}(T)] \leq \mathcal{O}\left(d\sqrt{(1 + V_T)T}\right).$$

### C.3 PROOF OF THEOREM 7

Define a sequence of random variable $\{Y_t\}_{t=1}^{\infty}$ as

$$Y_0 = 0 \quad \text{and} \quad Y_{t+1} = Y_t + \|\nabla f_t(x_t)\|^2 + \frac{f_t(x_{t+1}) - f_t(x_t)}{\eta} - \frac{L_1 L_0 d \delta}{2} - \frac{L_1 \eta}{2} \|\widehat{\nabla} f_t(x_t)\|^2.$$

Define the filtrations $\mathcal{F}_t = \sigma(x_1, \ldots, x_t)$. By eq. (28), we have $\{Y_t\}$ is a super-martingale. Moreover, we have

$$|Y_{t+1} - Y_t| = \left| \|\nabla f_t(x_t)\|^2 + \frac{f_t(x_{t+1}) - f_t(x_t)}{\eta} - \frac{L_1 L_0 d \delta}{2} - \frac{L_1 \eta}{2} \|\widehat{\nabla} f_t(x_t)\|^2 \right|$$

$$\overset{(i)}{\leq} \left( L_1^2 + L_0 \|\widehat{\nabla} f_t(x_t)\| + \frac{L_1 L_0 d \delta}{2} + \frac{L_1 \eta}{2} \|\widehat{\nabla} f_t(x_t)\|^2 \right) := c_t.$$

where $(i)$ follows because $\|\nabla f_t(x_t)\| \leq L_0$ and

$$f_t(x_{t+1}) - f_t(x_t) \leq L_0 \|x_{t+1} - x_t\| = \eta L_0 \|\widehat{\nabla} f_t(x_t)\|.$$

Applying the Azuma–Hoeffding inequality to $Y_{T+1}$, we have, for every $\epsilon > 0$,

$$\Pr(Y_{T+1} \geq \epsilon) \leq 2 \exp\left(-\frac{\epsilon^2}{\sum_{t=1}^{T} c_t^2}\right). \quad (32)$$

We next provide an upper bound on $\sum_{t=1}^{T} c_t^2$.

$$\sum_{t=1}^{T} c_t^2 = \sum_{t=1}^{T} \left( L_1^2 + L_0 \|\widehat{\nabla} f_t(x_t)\| + \frac{L_1 L_0 d \delta}{2} + \frac{L_1 \eta}{2} \|\widehat{\nabla} f_t(x_t)\|^2 \right)^2$$

$$\overset{(i)}{\leq} 4L_1^4 T + 4L_0^2 \sum_{t=1}^{T} \|\widehat{\nabla} f_t(x_t)\|^2 + L_1^2 L_0^2 d^2 \delta^2 T + L_1^2 \eta^2 \sum_{t=1}^{T} \left( \|\widehat{\nabla} f_t(x_t)\|^2 \right)^2$$

$$\overset{(ii)}{\leq} 4L_1^4 T + 4L_0^2 \sum_{t=1}^{T} \|\widehat{\nabla} f_t(x_t)\|^2 + L_1^2 L_0^2 d^2 \delta^2 T + L_1^2 \eta^2 \left( \sum_{t=1}^{T} \|\widehat{\nabla} f_t(x_t)\|^2 \right)^2, \quad (33)$$

where $(i)$ follows by applying $(a + b + c + d)^2 \leq 4a^2 + 4b^2 + 4c^2 + 4d^2$ to every term in the summation, and $(ii)$ follows because $\sum_{i=1}^{n} x_i^2 \leq (\sum_{i=1}^{n} x_i)^2$ when $x_i$'s are all positive.

Applying Lemma 1, we have

$$\sum_{t=1}^{T}\left\|\widehat{\nabla}f_t(x_t)\right\|^2 \leq \left\|\widehat{\nabla}f_1(x_1)\right\|^2\left(\sum_{t=1}^{T}\frac{1}{2^{t-1}}\right) + \frac{4d^2}{\delta^2}\sum_{t=1}^{T}\sum_{\tau=2}^{t}\frac{(f_\tau(x_\tau) - f_{\tau-1}(x_\tau))^2}{2^{t-\tau}} + 16d^2L_0^2T$$

$$\leq \frac{4M^2d^2}{\delta^2} + \frac{4d^2}{\delta^2}\sum_{t=1}^{T}(f_\tau(x_\tau) - f_{\tau-1}(x_\tau))^2\left(\sum_{\tau=2}^{t}\frac{1}{2^{t-\tau}}\right) + 16d^2L_0^2T$$

$$\leq \frac{4M^2d^2}{\delta^2} + \frac{8d^2MV_T}{\delta^2} + 16d^2L_0^2T$$

$$\overset{(i)}{\leq} 16d^2(L_0^2 + M^2)T, \tag{34}$$

where $(i)$ follows from the facts that $\delta = \sqrt{\frac{1+V_T}{T}}$ and $V_T \leq MT$.

Substituting eq. (34) into eq. (33), we obtain

$$\sum_{t=1}^{T}c_t^2 \leq (4L_1^4 + 64L_0^2d^2(L_0^2 + M^2))T + L_1^2L_0^2(1 + V_T) + \frac{16L_1^2}{L_0^2}d^2(L_0^2 + M^2)^2T(1 + V_T).$$

Therefore, there exist a constant $C > 0$, such that

$$\sum_{t=1}^{T}c_t^2 \leq Cd^2T(1 + V_T). \tag{35}$$

Substituting eq. (35) into eq. (32) and taking $\epsilon = \sqrt{Cd^2T(1 + V_T)\log\left(\frac{2}{\xi}\right)}$, we have, with probability greater or equal to $1 - \xi$

$$Y_{T+1} \leq d\sqrt{CT(1 + V_T)\log\left(\frac{2}{\xi}\right)}. \tag{36}$$

Substituting the definition of $Y_{T+1}$ into eq. (36) and rearranging, we obtain, with probability greater or equal to $1 - \xi$

$$\mathfrak{R}(T) = \sum_{t=1}^{T}\|\nabla f_t(x_t)\|^2$$

$$\leq \frac{2M}{\eta} + \frac{V_T}{\eta} + \frac{L_1L_0d\delta T}{2} + \frac{4L_1d^2M^2\eta}{\delta^2} + 8L_1L_0^2\eta d^2T + \frac{8ML_1\eta d^2V_T}{\delta^2}$$

$$+ d\sqrt{CT(1 + V_T)\log\left(\frac{2}{\xi}\right)}$$

$$= \mathcal{O}\left(d\sqrt{T(1 + V_T)\log\left(\frac{1}{\xi}\right)}\right). \tag{37}$$

## D  CONVENTIONAL ONE-POINT AND TWO-POINT GRADIENT ESTIMATORS

In this section, we introduce the conventional one-point and two-point gradient estimators, and provide the corresponding regret bounds if these estimators are adopted by OGD. These results serve as comparison to the new running difference gradient estimator that we propose in Section 5.

### D.1  REGRET OF CONVENTIONAL ONE-POINT GRADIENT ESTIMATOR

Consider the following one-point gradient estimator (Flaxman et al., 2005), which has been used in online convex optimization and is also widely adopted in offline gradient-free algorithms.

$$\widetilde{\nabla}f_t(x_t) := \frac{d}{\delta}f_t(x_t + \delta u_t)u_t, \tag{38}$$

where $u_t$ is drawn independently from the uniform distribution over the unit sphere in $\mathbb{R}^d$. It has been shown that

$$\mathbb{E}\left[\widetilde{\nabla} f_t(x_t)\right] = \nabla f_{t,\delta}(x_t),$$

where $f_{t,\delta}(x) = \mathbb{E}_u[f_t(x + \delta u)]$ with $u$ be uniformly distributed on the unit sphere. Although the expected value of such a one-point gradient estimator is close to the true gradient, i.e.,

$$\|\mathbb{E}[\widetilde{\nabla} f_t(x)] - \nabla f_t(x)\| = \|\nabla f_{t,\delta}(x) - \nabla f_t(x)\| \le L_1 \delta,$$

its variance explodes in the order of $\mathcal{O}(\frac{1}{\delta^2})$. The trade-off between the bias and variance prohibits us to choose a small $\delta$, and hence as we show below that the regret of OGD with such a one-point gradient estimator can be large.

Consider the OGD algorithm with the above one-point gradient estimator as described in Algorithm 4. The regret of such an algorithm is characterized in the following theorem.

---

**Algorithm 4** OGD with Conventional One-point Gradient Estimator

---

**Input:** Initial point $x_1$, stepsizes $\eta$ and parameter $\delta$
**for** $t = 1, \ldots, T$ **do**
  Draw $u_t$ from the uniform distribution over the unit sphere
  Observe $f_t(x_t + \delta u_t)$ and let $\widetilde{\nabla} f_t(x_t) = \frac{d u_t}{\delta} \cdot f_t(x_t + \delta u_t)$
  Let $x_{t+1} = x_t - \eta \widetilde{\nabla} f_t(x_t)$
**end for**

---

**Theorem 8.** *Suppose Assumption 1 holds. Consider Algorithm 4 with $x_0 = 0$, perturbation $\delta = \left(\frac{1+V_T}{T}\right)^{\frac{1}{4}}$ and stepsize $\eta = \frac{1}{d}\left(\frac{1+V_T}{T}\right)^{\frac{3}{4}}$. Then, the expected local regret satisfies*

$$\mathbb{E}\left[\mathfrak{R}(T)\right] \le \mathcal{O}(d(1 + V_T)^{\frac{1}{4}} T^{\frac{3}{4}}).$$

Clearly, the regret of OGD using the conventional one-point gradient estimator is much larger than that of OGD using our running difference gradient estimator given in Theorem 6 (note that $V_T$ scales at most with the order of $T$, and can scale much slower than $T$). The main reason is because the conventional one-point estimator has a much larger variance and hence the tradeoff between the bias and the variance requires a much large perturbation parameter $\delta$ which is chosen to be $\left(\frac{1+V_T}{T}\right)^{\frac{1}{4}}$. This consequently leads to a worse regret than our one-point estimator.

*Proof of Theorem 8.* By the Lipschitz-smooth condition, we have

$$f_t(x_{t+1}) \le f_t(x_t) + \langle \nabla f_t(x_t), x_{t+1} - x_t \rangle + \frac{L_1}{2}\|x_{t+1} - x_t\|^2$$

$$= f_t(x_t) - \eta \langle \nabla f_t(x_t), \widetilde{\nabla} f_t(x_t) \rangle + \frac{L_1 \eta^2}{2}\|\widetilde{\nabla} f_t(x_t)\|^2$$

$$\le f_t(x_t) - \eta \|\nabla f_t(x_t)\|^2 + \eta \langle \nabla f_t(x_t), \nabla f_t(x_t) - \widetilde{\nabla} f_t(x_t) \rangle + \frac{L_1 d^2 \eta^2 M^2}{2\delta^2}. \quad (39)$$

Rearranging and taking expectation on both sides of eq. (39) conditioned on $\mathcal{F}_t = \sigma(u_1, \ldots, u_{t-1})$, we obtain

$$\|\nabla f_t(x_t)\|^2 \overset{(i)}{\le} \frac{f_t(x_t) - \mathbb{E}\left[f_t(x_{t+1})|\mathcal{F}_t\right]}{\eta} + \langle \nabla f_t(x_t), \nabla f_t(x_t) - \nabla f_{t,\delta}(x_t) \rangle + \frac{L_1 d^2 M^2 \eta}{2\delta^2}$$

$$\overset{(ii)}{\le} \frac{f_t(x_t) - \mathbb{E}\left[f_t(x_{t+1})|\mathcal{F}_t\right]}{\eta} + \frac{L_0 L_1 d\delta}{2} + \frac{L_1 d^2 M^2 \eta}{2\delta^2}, \quad (40)$$

where $(i)$ follows from the steps similar to those in the proof of Lemma 2 and $(ii)$ follows from the Cauchy-Schwartz inequality, and because $\|\nabla f_t(x_t)\| \le L_0$, and $\|\nabla f_t(x_t) - \nabla f_{t,\delta}(x_t)\| \le L_1 \delta$.

Taking expectation on both sides of eq. (40) and telescoping, we obtain

$$\mathbb{E}[\mathfrak{R}(T)]$$

$$= \sum_{t=1}^{T} \mathbb{E}\left[\|\nabla f_t(x_t)\|^2\right]$$

$$\leq \frac{f_1(x_1) - \mathbb{E}\left[f_{T+1}(x_{T+1})\right]}{\eta} + \frac{1}{\eta}\sum_{t=1}^{T}\mathbb{E}\left[f_{t+1}(x_{t+1}) - f_t(x_t)\right] + \frac{L_0 L_1 d\delta T}{2} + \frac{L_1 d^2 M^2 \eta T}{2\delta^2}$$

$$\leq \frac{2M + V_T}{\eta} + \frac{L_0 L_1 d\delta T}{2} + \frac{L_1 d^2 M^2 \eta T}{2\delta^2}$$

$$\leq \mathcal{O}(d(1 + V_T)^{\frac{1}{4}} T^{\frac{3}{4}}).$$

$\square$

## D.2 REGRET OF TWO-POINT GRADIENT ESTIMATOR

Although the two-point gradient estimator is not applicable to our setting with only a single function value feedback, it serves as a desirable comparison benchmark for our designed one-point estimator. In the literature, the standard two-point gradient estimators have been shown to achieve much better performance than the conventional one-point estimator as well as a recently developed one-point estimator in Zhang et al. (2022) in both online and offline optimization. In this section, by establishing the regret of OGD with the standard two-point estimator, we will show that our running difference one-point estimator can achieve the same performance as the standard two-point estimator.

Consider the standard two-point gradient estimator (Agarwal et al., 2010) given as follows:

$$\bar{\nabla} f_t(x_t) := \frac{d}{\delta}(f_t(x_t + \delta u_t) - f_t(x_t))u_t. \tag{41}$$

Compared to eq. (38), the two-point estimator in eq. (41) has the same expected value, but its variance is bounded by $L_0^2 d^2$, which is much smaller.

Now consider OGD with the above two-point gradient estimator as described in Algorithm 5. The regret of such an algorithm is characterized in the following theorem.

---

**Algorithm 5** OGD with Standard Two-point Gradient Estimator

---

**Input:** Initial point $x_1$, stepsizes $\eta$ and parameter $\delta$
**for** $t = 1, \ldots, T$ **do**
    Draw $u_t$ from the uniform distribution over the unit sphere
    Observe $f_t(x_t + \delta u_t)$ and $f_t(x_t)$
    Let $\bar{\nabla} f_t(x_t) = \frac{d u_t}{\delta} \cdot (f_t(x_t + \delta u_t) - f_t(x_t))$
    Let $x_{t+1} = x_t - \eta \bar{\nabla} f_t(x_t)$
**end for**

---

**Theorem 9.** *Suppose Assumption 1 holds. Consider Algorithm 5 with $x_0 = 0$, perturbation $\delta = \sqrt{\frac{1+V_T}{T}}$ and stepsize $\eta = \frac{1}{d}\sqrt{\frac{1+V_T}{T}}$. Then, the expected local regret satisfies*

$$\mathbb{E}\left[\mathfrak{R}(T)\right] \leq \mathcal{O}(d\sqrt{(1 + V_T)T}).$$

Clearly, comparison of Theorem 6 and Theorem 9 shows that our one-point estimator achieves the same regret as the two-point estimator, and they both outperform the conventional one-point estimator. This is the first result in online nonconvex optimization that establishes that one function value oracle achieves the same regret as two oracles.

*Proof of Theorem 9.* By the Lipschitz-smooth condition, we have

$$f_t(x_{t+1}) \leq f_t(x_t) + \langle \nabla f_t(x_t), x_{t+1} - x_t \rangle + \frac{L_1}{2}\|x_{t+1} - x_t\|^2$$

$$= f_t(x_t) - \eta \langle \nabla f_t(x_t), \bar{\nabla} f_t(x_t) \rangle + \frac{L_1 \eta^2}{2}\|\bar{\nabla} f_t(x_t)\|^2$$

$$\leq f_t(x_t) - \eta \|\nabla f_t(x_t)\|^2 + \eta \langle \nabla f_t(x_t), \nabla f_t(x_t) - \bar{\nabla} f_t(x_t) \rangle + \frac{L_1 L_0 d^2 \eta^2}{2}. \quad (42)$$

Rearranging and taking expectation on both sides of eq. (42) conditioned on $\mathcal{F}_t = \sigma(u_1, \ldots, u_{t-1})$, we obtain

$$\|\nabla f_t(x_t)\|^2 \leq \frac{f_t(x_t) - \mathbb{E}\left[f_t(x_{t+1})|\mathcal{F}_t\right]}{\eta} + \langle \nabla f_t(x_t), \nabla f_t(x_t) - \nabla f_{t,\delta}(x_t) \rangle + \frac{L_1 L_0 d^2 \eta}{2}$$

$$\overset{(i)}{\leq} \frac{f_t(x_t) - \mathbb{E}\left[f_t(x_{t+1})|\mathcal{F}_t\right]}{\eta} + \frac{L_0 L_1 d\delta}{2} + \frac{L_1 L_0 d^2 \eta}{2}, \quad (43)$$

where $(i)$ follows from the steps similar to those in the proof of Lemma 2 and Cauchy-Schwartz inquality.

Taking expectation on both sides of eq. (43) and telescoping, we obtain

$$\mathbb{E}[\mathfrak{R}(T)]$$

$$= \sum_{t=1}^{T} \mathbb{E}\left[\|\nabla f_t(x_t)\|^2\right]$$

$$\leq \frac{f_1(x_1) - \mathbb{E}\left[f_{T+1}(x_{T+1})\right]}{\eta} + \frac{1}{\eta} \sum_{t=1}^{T} \mathbb{E}\left[f_{t+1}(x_{t+1}) - f_t(x_t)\right] + \frac{L_0 L_1 d\delta T}{2} + \frac{L_1 L_0^2 d^2 \eta T}{2}$$

$$\leq \mathcal{O}(d\sqrt{T(1 + V_T)}).$$

$\square$

