# OpenReview forum: "On the Hardness of Online Nonconvex Optimization with Single Oracle Feedback"
_ICLR.cc/2024/Conference — ICLR 2024 poster_

### Official Review · Reviewer_e2ZJ · 2023-10-21

**Soundness:** 3 good
**Presentation:** 4 excellent
**Contribution:** 3 good
**Rating:** 6
**Confidence:** 4

**Summary:**

This work studies the problem of online nonconvex optimization with single oracle feedback. More specifically, the authors consider three setups of the single oracle: single exact gradient feedback, single stochastic gradient feedback, and single function value feedback. The authors choose local regret on the original function, not on the smoothed version of it as previous works, as the performance measure. The key contribution of this work is that the authors find that the problem-dependent quantity of function variation is a fundamental variable in this problem. Specifically, in the first two cases, the authors give two corresponding lower bounds regarding the function variation for the linear-span algorithm family. And they prove that OGD and OSGD are optimal inside the linear-span algorithm family. In the functional value feedback, the authors use a new running difference gradient estimator and obtain an optimal result inside the linear-span algorithm family. Note that the result under the function value feedback with one-point running difference gradient estimator is as good as that using a two-point gradient estimator.

**Strengths:**

The overall paper is well-written. The problem is well-motivated, and the descriptions of the three cases are well-explained with proof sketches. The observation of the role of function variations in this problem is important and may lead to novel perspectives for this problem for future research.

**Weaknesses:**

I have only one major concern, which affects my score for this paper.

Concretely, I think some statements of the optimality of the results are a little over-claimed. Note that the lower bounds (Theorem 1 and Theorem 3) do not hold for any possible algorithm but only for a specific family called linear-span algorithms. Although the family of linear-span algorithms contains many famous and widely used algorithms such as GD, AGD, and so on, as the authors have stated, they cannot represent all algorithms. As a result, the lower bounds are only algorithm-related lower bounds. When talking about optimality, it is correct to say that 'our algorithm is optimal inside the linear-span algorithm', but not just 'our algorithm is optimal', as the authors have claimed in the current version. This is a serious issue, and I suggest that the authors could revise the corresponding statements to avoid unnecessary misunderstandings of the paper's contributions. By the way, I think the contributions of this paper are still adequate for acceptance, even if the authors use the correct descriptions of optimality. As a result, there is no need for over-claiming, which will only give readers and reviewers (at least for me) a bad impression.

**Questions:**

1. Can I conclude that after this work, the methods using the smoothed function $F_{t,w}$ can be abandoned? Or equivalently, does this work offer a strictly superior methodology compared with Hazan's work? If not, what is the disadvantages of measuring local regret defined on the original functions?
2. Is the one-point running difference gradient estimator strictly superior to the standard FKM one-gradient estimator? I think this only holds in the problem of online nonconvex optimization, but not in bandit convex optimization (BCO)? As the authors have stated, the variance of the one-point running difference estimator is a constant, which is also the case for the standard two-point estimator, and thus the guarantee of this work is as good as that using a two-point estimator. However, is the variance of the running difference estimator still a constant in the BCO setup? If so, it seems that an $O(\sqrt{T})$ regret can be achieved in the BCO setup, which is definitely the most exciting progress in the BCO research (which is too simple to be true). If not, what is the difference between using this estimator in the nonconvex setup and in the convex setup? Will using the running difference estimator in the method of Flaxman (SODA'05) improve their $O(T^{3/4})$ regret? If not, what is the key difficulty?

---

> ### Author Response · Authors · 2023-11-20
>
> We thank the reviewer for the careful reading and thoughtful comments. We addressed the reviewer's questions in the following and revised the paper accordingly. The changes are marked in blue color in our revision. We hope the responses below and the changes in the paper address the reviewer's concerns.
>
> **Q1:** Concretely, I think some statements of the optimality of the results are a little over-claimed. Note that the lower bounds (Theorem 1 and Theorem 3) do not hold for any possible algorithm but only for a specific class called linear-span algorithms. When talking about optimality, it is correct to say that 'our algorithm is optimal inside the class of linear-span algorithms', but not just 'our algorithm is optimal', as the authors have claimed in the current version. This is a serious issue, and I suggest that the authors could revise the corresponding statements to avoid unnecessary misunderstandings of the paper's contributions. By the way, I think the contributions of this paper are still adequate for acceptance, even if the authors use the correct descriptions of optimality. As a result, there is no need for over-claiming, which will only give readers and reviewers (at least for me) a bad impression.
>
> **A1:** Many thanks for pointing this out! We agree with the reviewer that our claims of optimality should be more precise and restricted to the class of linear-span algorithms. Following the reviewer's suggestion, we have updated all these claims in the revision. Thanks again for the suggestion, and we agree that it is important to clarify our contribution.
>
> **Q2:** Can I conclude that after this work, the methods using the smoothed function  $F_{t,w}$ can be abandoned? Or equivalently, does this work offer a strictly superior methodology compared with Hazan's work? If not, what is the disadvantages of measuring local regret defined on the original functions?
>
> **A2:**  We think that the previous studies (e.g., Hazan et al., 2017) investigated a **complimentary** regret metric based on the window-smoothed $F_{t,w}$, which is also valuable, particularly for studying slowly changing environments. In the future, it is also interesting to continue their line of work and obtain refined problem-dependent bounds relating to function variation $V_T$.
>
> **Q3:** Is the one-point running difference gradient estimator strictly superior to the standard FKM one-gradient estimator? I think this only holds in the problem of online nonconvex optimization, but not in bandit convex optimization (BCO)? As the authors have stated, the variance of the one-point running difference estimator is a constant, which is also the case for the standard two-point estimator, and thus the guarantee of this work is as good as that using a two-point estimator. However, is the variance of the running difference estimator still a constant in the BCO setup? If so, it seems that an $\mathcal{O}(\sqrt{T})$ regret can be achieved in the BCO setup, which is definitely the most exciting progress in the BCO research (which is too simple to be true). If not, what is the difference between using this estimator in the nonconvex setup and in the convex setup? Will using the running difference estimator in the method of Flaxman (SODA'05) improve their $\mathcal{O}(T^{3/4})$  regret? If not, what is the key difficulty?
>
> **A3:** Many thanks for the insightful questions! Yes, the one-point running difference gradient estimator is strictly better than FKM in our setting (i.e., online nonconvex optimization). But this may not generally hold, e.g., for online BCO. In particular, for BCO, the variance of the running difference estimator may not be a constant if there is no assumption on the function variation $V_T$. Hence, $\mathcal{O}(\sqrt{T})$ may not be achieved without function variation $V_T$ being present in the upper bound.
>
> We also point out that (Flaxman et al., SODA'2005) considered BCO for **static** global regret and did not introduce function variation $V_T$ in their static setting. Hence, our analysis of the running difference estimator for **dynamic** regret does not apply to their setting directly. However, if we consider BCO under **dynamic regret**, our method will be applicable, and the bound will involve the function variation $V_T$.
>
> Finally, we thank the reviewer again for the helpful comments and suggestions for our work. The reviewer's main concern is our claim of the optimality of algorithms, which we have revised our statement throughout the paper. Since the reviewer commented that "I think the contributions of this paper are still adequate for acceptance", we kindly ask the reviewer to consider raising the rating of our work to be aligned with your comment. Certainly, we are more than happy to address any further questions that you may have during the discussion period.

---

> > ### Comment · Reviewer_e2ZJ · 2023-11-20
> > **Response to Authors**
> >
> > Thanks for the detailed feedback. Since my main concerns about the overclaims on the optimality of the algorithms are solved. I am happy to update the score of this paper and recommend acceptance. Besides, I suggest that the authors could stress more about the relationship with [Hazan et al., 2017], as explained in A2.

---

> > > ### Author Response · Authors · 2023-11-20
> > >
> > > We thank the reviewer for the prompt response and the further suggestion. We have included the discussion in A2 to Section 1.2 under "Online nonconvex optimization" in the newest revision in blue color. Please feel free to check.

---

### Official Review · Reviewer_qRqP · 2023-10-25

**Soundness:** 3 good
**Presentation:** 3 good
**Contribution:** 3 good
**Rating:** 6
**Confidence:** 3

**Summary:**

This paper studies the online nonconvex optimization problem, where access to only a single oracle is allowed per time step. The authors consider three variants: single gradient oracle (SGO) feedback, single stochastic gradient oracle (SSGO) feedback, and single value oracle (SVO) feedback, and take the local regret of the original objective functions as the performance metric. For SGO and SSGO, they derive lower bounds on the local regret and show that the classic online algorithms are already optimal. For SVO, they develop an online algorithm based on a one-point running difference gradient estimator, which achieves a local regret that a generic stochastic gradient oracle can best achieve.

**Strengths:**

- The online nonconvex optimization problem is interesting and relevant.
- The three proposed variants are practical and meaningful.
- The theoretical results are standard, specifically, indicating the importance of parameter $V_T$.

**Weaknesses:**

- I would like to see more comparisons and discussions on three variants, specifically, when we need to consider SGO/SSGO/SVO.
- There is no matching lower bound for the SVO problem.

**Questions:**

- Can the lower bound analysis extend to the general setting of sliding windows? E.g., for a fixed window length $w$, is $V_T$ still appear in the regret bound?

---

> ### Author Response · Authors · 2023-11-20
>
> We thank the reviewer for the careful reading and thoughtful comments. We addressed the reviewer's questions in the following and revised the paper accordingly. The changes are marked in blue color in our revision. We hope the responses below and the changes in the paper address the reviewer's concerns.
>
> **Q1:** I would like to see more comparisons and discussions on three variants, specifically when we need to consider SGO/SSGO/SVO.
>
> **A1:** Thanks for pointing this out. In the revision, we added the discussion in Section 1.1 (see the blue-colored texts). We also provide the discussion below for your convenience. The three settings correspond to three different application scenarios. The SGO setting applies to white-box systems with known objective functions whose exact gradients can be calculated. However, many systems can have various uncertainties, e.g., loss functions depending on random samples or systems with intrinsic noise. In such a case, even if the system objective function is known, the calculation of their gradients can still be stochastic, leading to the SSGO setting. Moreover, the SVO setting is suitable to model black-box systems, where objective functions are unknown but only function values can be queried. For example, the recommendation system can provide users' ratings (i.e., reward values), but the reward function following which users provide ratings is typically unknown.
>
> **Q2:** There is no matching lower bound for the SVO problem.
>
> **A2:** Yes, it is still an open problem. We anticipate the lower bound for SVO to be $\mathcal{O}(d\sqrt{(1+V_T)T})$, but this seems to need non-trivial techniques to prove. Here are some initial thoughts. Inspired by (Besbes et al., 2015) on the lower bound for online convex optimization with function value feedback, we will need to construct several special nonconvex functions as candidates for $f_t$'s and then reformulate the problem on $f_t$ as a non-parametric estimation problem, aiming to identify which function is used from the candidate list at each time step. We note that constructing nonconvex candidate functions and reformulating the problem will require considerable effort, and we leave this as an interesting future problem.
>
> **Q3:** Can the lower bound analysis extend to the general setting of sliding windows? E.g., for a fixed window length $w$, $V_T$ is still appear in the regret bound?
>
> **A3:** Good question. We expect that our lower bound analysis can be extended to the setting of sliding windows, but may not be in a straightforward way. In our lower bound proof, we construct functions in different blocks to be orthogonal with each other. After sliding window average, functions may not satisfy such a requirement anymore. To address this issue, setting a window of initial and ending functions to be zero in each block can potentially avoid such a problem. It may be possible to design better schemes, but will take more effort. We will continue to explore this problem in the future.
>
>
> (Besbes et.al., 2015): Besbes, Omar, Yonatan Gur, and Assaf Zeevi. "Non-stationary stochastic optimization." Operations research 63, no. 5 (2015): 1227-1244.

---

### Official Review · Reviewer_sA6m · 2023-10-25

**Soundness:** 3 good
**Presentation:** 3 good
**Contribution:** 4 excellent
**Rating:** 8
**Confidence:** 3

**Summary:**

This paper studies online-nonconvex-optimization (ONO) problem with local regret. Unlike previous works that focused on window-smoothed loss of the form $F_t(x)=\frac{1}{w}\sum\_{i=t-w+1}^t f_i(x)$, this paper directly addresses local regret using the original losses, namely $\mathrm{Regret}\_T = \sum\_{t=1}^T \\|\nabla f_t(x_t)\\|^2$ (which corresponds to setting $w=1$ in the window-smoothed definition). Moreover, the paper focuses on single oracle algorithm where, in each iteration, the algorithm can only request one oracle query. Specifically, three oracle models are analyzed in this work: 1. deterministic gradient oracle (SGO) that returns $\nabla f_t(x_t)$, 2. stochastic gradient oracle (SSGO) that returns an unbiased estimator of $\nabla f_t(x_t)$, and 3. function value oracle (SVO) that returns $f_t(x_t)$. For the first two oracle models, the paper presents a tight analysis, providing both upper and lower bounds. For the third model, the paper offers improvements over existing state-of-the-art results.

**Strengths:**

This paper offers a thorough examination of the ONO using single-oracle algorithms. I'd like to highlight a few standout results:

- While previous results of ONO with local regret consider window-smoothed approximation $F_t(x)=\frac{1}{w}\sum\_{i=t-w+1}^t f_i(x)$, which may vary a lot from the original loss, this paper works directly on original losses and thus providing a more accurate regret bound.

- The algorithms studied in this paper only makes one single oracle query in each iteration, while most algorithms in prior works require multiple oracle queries due to window-smoothing.

- This paper provides comprehensive analysis for 3 different oracle models. Specifically, it provides lower bounds for the gradient oracle models and proves that online subgradient descent achieves the optimal lower bounds in both cases.

  Moreover, for the function value model, this paper proposes a single oracle algorithm, matching the state-of-art regret bound which is previously achieved by an algorithm making two oracle queries per iteration. The idea of one-point estimator using running difference of function values instead of the standard two-point estimator is novel and interesting.

- What stands out to me the most is the notation of "function variation over time", namely $V_T = \sum_{t=2}^T \sup_x |f_t(x) - f_{t-1}(x)|$. Similar to the notation of path length in dynamic regret, function variation is also an adaptive measure that captures the difficulty of the ONO problem. Therefore, the proposed regret bounds in this paper (namely $O(1+V_T)$ for SGO, $O(\sqrt{(1+V_T)T})$ for SSGO, and $O(d\sqrt{(1+V_T)T})$ for SVO) are always tighter than the previous vacuous bounds in the corresponding settings.

  I also find it interesting how $V_T$ naturally follows from $\sum\_{t=1}^T f_t(x_t) - f_t(x_{t-1})$ in the smooth loss analysis, and it just happens to be a tighter measure.

**Weaknesses:**

The lower bounds for SGO and SSGO models are restricted to a relatively small family of algorithms such that $x_{t+1} = x_1 + \sum\_{i=1}^t a_{t,i} g_i$. This family does not include popular algorithms like online mirror descent and FTRL.

**Questions:**

1. For the lower bounds, I assume there is an implicit assumption on function dimension $d$ (where $f_t:\mathbb{R}^d\to\mathbb{R}$), e.g. $d\ge \Omega(1+V_T)$ to guarantee there are $\Omega(1+V_T)$ orthogonal gradients?

2. The lower bound construction in Carmon et al. 2021 and Arjevani et al. 2022 can be applied to a larger family of algorithms besides algorithms of form $x_{t+1} = x_1 + \sum\_{i=1}^t a_{t,i} g_i$. I am curious if the lower bounds for ONO can also be extended beyond the current family? If not, what is the main difficulty in ONO compared to offline nonconvex optimization?

3. proof of Lemma 2 first two equalities: where does $d$ out side $\\|\mathbb{E}...\\|$ come from? is it a typo?

---

> ### Author Response · Authors · 2023-11-20
>
> We thank the reviewer for the careful reading and thoughtful comments. We addressed the reviewer's questions in the following and revised the paper accordingly. The changes are marked in blue color in our revision. We hope the responses below and the changes in the paper address the reviewer's concerns.
>
> **Q1:** The lower bounds for SGO and SSGO models are restricted to a relatively small family of algorithms such that $x_{t+1} = x_t+ \sum_{i=1}^t\alpha_{t,i}g_i$. This family does not include popular algorithms like online mirror descent and FTRL.
>
> **A1:** Thanks for the great comment!
> Our current technique can extend to the larger family of algorithms of the zero-respecting set (as discussed in A3). To develop lower bounds for the larger algorithm class that includes mirror descent and FTRL, the general idea for online convex optimization counterpart may be useful. But the construction of specific hard-to-learn nonconvex functions will take more effort. We will pursue this direction as an interesting future work.
>
> **Q2:** For the lower bounds, I assume there is an implicit assumption on function dimension $d$ (where $f_t:\mathbb{R}^d \to \mathbb{R}$), e.g. $d\ge \Omega(1+V_T)$  to guarantee there are $\Omega(1+V_T)$ orthogonal gradients?
>
> **A2:** We agree with the reviewer and made this assumption explicit in the revision of the paper.
>
> **Q3:** The lower bound construction in Carmon et al. 2021 and Arjevani et al. 2022 can be applied to a larger family of algorithms besides algorithms of form $x_{t+1} = x_t+ \sum_{i=1}^t\alpha_{t,i}g_i$. I am curious if the lower bounds for ONO can also be extended beyond the current family? If not, what is the main difficulty in ONO compared to offline nonconvex optimization?
>
> **A3:** Great question! We expect our proof of lower bound can be extended to the suggested larger family of algorithms, i.e., all zero-respecting algorithms (Carmon et al., 2021; Arjevani et al., 2022). Specifically, to make the current proof still hold for the zero-respecting algorithm class, the gradients in the previous blocks should be orthogonal to the present gradients. Our current design of online functions already satisfies such a requirement. Hence, our lower bound will still hold. We will carefully go over the proof and ensure that all steps are mathematically rigorous.
>
> **Q4:** Proof of Lemma 2 first two equalities: where does $d$ outside $\\|\mathbb{E}\ldots\\|$ come from? is it a typo?
>
> **A4:** Thanks for pointing it out. It is a typo, and $d$ should come out later. We corrected the steps in the proof of Lemma 2 in the revised paper (see the highlighted steps in the proof of Lemma 2).

---

> > ### Comment · Reviewer_sA6m · 2023-11-20
> > **Response to Authors**
> >
> > Thank you for the detailed response and updated revision. It will be very interesting to see a formal lower bound proof for the more general family of ONO algorithms in terms of function variation.

---

> > > ### Author Response · Authors · 2023-11-21
> > >
> > > We thank the reviewer for the prompt response. Yes, the lower bound for the more general family of algorithms is intriguing, and we will continue working on this.

---

### Official Review · Reviewer_xXV1 · 2023-11-02

**Soundness:** 3 good
**Presentation:** 3 good
**Contribution:** 3 good
**Rating:** 6
**Confidence:** 3

**Summary:**

This paper studies minimizing the local regret in online non-convex optimization. Previous work mainly focus on smoothed version of regret, and show the **worst-case** local regret with window size 1 is linear with respect to T. In this paper, the authors reveal that, both upper bound and lower bound in this setting can be better, as they can depend on the total function variation. The authors consider several different settings in ONCO with different oracles, such as sgo and ssgo, and show matching function-variation-dependent lower and upper bounds.

**Strengths:**

Significance and novelty:

The main idea is quite novel and very interesting. In recent several years, there have been a lot of work on studying the local regret. However, all of these work use smoothed version of the loss function (i.e., the average of a batch of functions in a window) when computing the regret, and people typically believed that using the exact function (that is, with window size 1) will lead to a \Omega(T) regret. However, these bounds are actually worst-case bound, and do not depend on data of the loss functions themselves. That is to say, there must exist simple cases where the regret is sublinear. In this paper, the authors captures this by showing a data-dependnt lower bound (function variation bound) and matching upper bound. This is the first paper that shows both sublinear lower and upper bounds for online non-convex optimization with local regret with window 1, which is a significant and novel contribution, and may inspire future works.


The proof for the upper bound is relatively straightforward and brings to mind what people do when obtaining the function variation  bound  for dynamic regret. The proof for lower bound is much more challenging and require novel techniques.

Presentation: The paper is in general well-written, and easy to read.

**Weaknesses:**

There is a long line of research in online learning that studies dynamic regret, which also have function variation bounds. After reading the proof, I found that the many parts of proof for obtaining the **upper** bound are very similar to this line of research. I recommend the authors review these papers and add a discussion.

**Questions:**

In online optimization, apart from function variation bound, there are also other measues for function changes, such as the gradient variation bound. Is it possible to also consider this or other adaptive bounds (in future work)? Similar to function variation, the gradient variation bound is defined as: V_T = \sum_{t=1}^T \sup_{x} ||\nabla f_t(x) - \nabla f_{t-1}(x)||^2. See, e.g., (1) of [1], for more information.

[1] Chiang, C. K., Yang, T., Lee, C. J., Mahdavi, M., Lu, C. J., Jin, R., & Zhu, S.. Online optimization with gradual variations. In Conference on Learning Theory, 2012.

> Introduction: One line of work adopted the global regret as the performance metric (Krichene et al., 2015; Agarwal et al., 2019; Lesage-Landry et al., 2020; Heliou et al., 2020), which compares the algorithm output to the global minimum of the nonconvex objective functions. However, accessing the global minimum of nonconvex functions is typically infeasible.

I don't understand this discussion. Regret is a performance metric, and people do not really compute regret when implementing the algorithm. So why "accessing the global minimum of nonconvex functions is typically infeasible"?

---

> ### Author Response · Authors · 2023-11-20
>
> We thank the reviewer for the careful reading and thoughtful comments. We addressed the reviewer's questions in the following and revised the paper accordingly. The changes are marked in blue color in our revision. We hope the responses below and the changes in the paper address the reviewer's concerns.
>
> **Q1:** There is a long line of research in online learning that studies dynamic regret, which also have function variation bounds. After reading the proof, I found that the many parts of proof for obtaining the upper bound are very similar to this line of research. I recommend the authors review these papers and add a discussion.
>
> **A1:** Thanks for the suggestion. In the revision, we have added the suggested discussion to Section 1.2 Related Work. Please feel free to check.
>
> **Q2:** In online optimization, apart from function variation bound, there are also other measures for function changes, such as the gradient variation bound. Is it possible to consider this or other adaptive bounds (in future work)? Similar to function variation, the gradient variation bound is defined as $V_T = \sum_{t=1}^T \sup_{x} ||\nabla f_t(x) - \nabla f_{t-1}(x)||^2$. See, e.g., (1) of (Chiang et al., 2012), for more information.
>
> **A2:** We appreciate your insightful inclusion of the $L_p$-deviation concept (Chiang et al., 2012) into this discussion. It is possible to derive upper bounds based on other types of function variation. Consider the example of the gradient variation suggested by the reviewer. Here are our preliminary thoughts. We can use a specific $f_t$ as the reference function, and then the updates at other time steps can be viewed as gradient descents of the reference function with a bias error, where the gradient variation determines the bias error. Such an idea may lead to a gradient variation-dependent dynamic regret upper bound. We will continue to work out the rigorous proof, which we expect to be promising.
>
> **Q3:** I don't understand this discussion (quoted texts in the introduction). Regret is a performance metric, and people do not really compute regret when implementing the algorithm. So why "accessing the global minimum of nonconvex functions is typically infeasible"?
>
> **A3:** Great question. Typically, the comparison metric in the regret serves as a baseline to measure how well-designed algorithms perform. Since the global minimum of nonconvex functions is generally infeasible to obtain in a computationally efficient manner, using such a point as the comparison baseline might be an unrealistic standard for computationally feasible algorithms to try to achieve.

---

### Meta-Review · Area_Chair_D3Hx · 2023-12-11

**Metareview:**

The paper under review advances the theoretical understanding of online non-convex optimization (ONO) and provides results on achievable local regret bounds using only a single oracle feedback. The reviewers all agree that the presented results are substantial contributions and the paper should be accepted.

**Justification For Why Not Higher Score:**

The paper meets the bar acceptance, but it is not clear whether it should be considered one of the top submissions.

**Justification For Why Not Lower Score:**

N/A

---

### Decision · Program_Chairs · 2024-01-16

Accept (poster)